# High p16 expression and heterozygous *RB1* loss are biomarkers for CDK4/6 inhibitor resistance in ER⁺ breast cancer

Marta Palafox[1], Laia Monserrat[1], Meritxell Bellet[2,3], Guillermo Villacampa[4], Abel Gonzalez-Perez [5,6], Mafalda Oliveira [2,3], Fara Brasó-Maristany[7], Nusaibah Ibrahimi [8,9], Srinivasaraghavan Kannan[10], Leonardo Mina[11], Maria Teresa Herrera-Abreu[12], Andreu Òdena[1], Mònica Sánchez-Guixé [1], Marta Capelán[2,3], Analía Azaro[2,3], Alejandra Bruna[13], Olga Rodríguez[1], Marta Guzmán[1], Judit Grueso[1], Cristina Viaplana[4], Javier Hernández [14], Faye Su[15], Kui Lin[16], Robert B. Clarke [17], Carlos Caldas [18], Joaquín Arribas [19,20,21,22,23], Stefan Michiels [8,9], Alicia García-Sanz[11], Nicholas C. Turner[12], Aleix Prat [7,24,25,26,27], Paolo Nuciforo [28], Rodrigo Dienstmann[4], Chandra S. Verma[10,29,30], Nuria Lopez-Bigas [5,6,23], Maurizio Scaltriti[31], Monica Arnedos[32,33], Cristina Saura [2,3] & Violeta Serra [1,19] ✉

CDK4/6 inhibitors combined with endocrine therapy have demonstrated higher antitumor activity than endocrine therapy alone for the treatment of advanced estrogen receptor-positive breast cancer. Some of these tumors are de novo resistant to CDK4/6 inhibitors and others develop acquired resistance. Here, we show that p16 overexpression is associated with reduced antitumor activity of CDK4/6 inhibitors in patient-derived xenografts (*n* = 37) and estrogen receptor-positive breast cancer cell lines, as well as reduced response of early and advanced breast cancer patients to CDK4/6 inhibitors (*n* = 89). We also identified heterozygous *RB1* loss as biomarker of acquired resistance and poor clinical outcome. Combination of the CDK4/6 inhibitor ribociclib with the PI3K inhibitor alpelisib showed antitumor activity in estrogen receptor-positive non-basal-like breast cancer patient-derived xenografts, independently of *PIK3CA*, *ESR1* or *RB1* mutation, also in drug de-escalation experiments or omitting endocrine therapy. Our results offer insights into predicting primary/acquired resistance to CDK4/6 inhibitors and post-progression therapeutic strategies.

The combination of cyclin D-dependent kinases 4 and 6 (CDK4/6) inhibitors (CDK4/6i) palbociclib, ribociclib and abemaciclib with endocrine therapy (ET) has been approved for the treatment of patients with advanced estrogen receptor-positive (ER⁺) and human epidermal growth factor receptor 2 (HER2)-negative breast cancer (BC)[1–4]. In the adjuvant setting, abemaciclib in combination with ET improves disease-free survival compared with ET alone[5]. Abemaciclib or palbociclib in combination with trastuzumab and ET has also shown clinical activity in HER2-positive (HER2⁺)/ER⁺ BC[6,7]. Despite the clinical success of these treatments in these subsets of BC, the identification of

---

biomarkers of response to CDK4/6i plus ET as well as designing therapeutic strategies for treating patients that escape from this therapy remains a major clinical need. Previous studies have highlighted that tumor sensitivity to CDK4/6 inhibition require a physiologically functional $G_1$-S restriction point, as well as the absence of mechanisms that activate the cyclin E/CDK2 complex[8–11]. More specifically, high CDK6 results in a reduced response to CDK4/6 inhibitors[12–14]. Amplification and overexpression of *FGFR1* has been associated with high expression of cyclin D1, resistance to antiestrogens alone and in combination with CDK4/6i[9]. Loss of pRb itself implies the complete loss of cell cycle regulation at the $G_1$-S restriction point for which CDK4/6i are no longer effective[15,16]. Alternatively, high cyclin E results in resistance to CDK4/6i, as it bypasses the requirement of CDK4/6 for cell cycle progression[17].

PI3K inhibitors (PI3Ki), in combination with fulvestrant, have been approved for the treatment of ER+ metastatic BC with *PIK3CA* mutation[18]. The activity of combining PI3Ki and ET (NCT03056755) as well as triple combinations of CDK4/6i, PI3Ki and ET is being investigated in patients whose tumors progress after CDK4/6i treatment (NCT01872260; NCT02088684; NCT03056755). Results from the PIPA trial (NCT02389842) have also shown a 37.5% response rate of palbociclib with the PI3Ki taselisib and fulvestrant in *PIK3CA*-mutant ER + BC[19].

In this study, we aimed to identify biomarkers of primary and acquired resistance to CDK4/6i in a panel of 37 patient-derived tumor models, using genetic, transcriptomic and proteomic approaches. Additionally, we explored if the combination of a PI3Ki plus a CDK4/6i has therapeutic potential in ER+ and in HER2+ BC with primary or acquired resistance to CDK4/6i, and its association with the *PIK3CA, ESR1, RB1*-mutation status.

## Results

### Ribociclib monotherapy has higher antitumor activity than other targeted agents in ER+ and HER2+ BC PDXs

Patient-derived xenografts (PDXs) are clinically relevant preclinical models for drug screening and biomarker identification[20,21]. We obtained 58 BC PDXs from implanting 473 ER+ BC tumor specimens in immune-deficient mice (12% of success rate; Fig. 1A). Among the established PDXs, 12 from primary tumors and 9 from metastatic biopsies were initially available for the study (Supplementary Tables 1, 2). Overall, the PDXs recapitulated the molecular subtype of the corresponding original tumor, with the exception of PDX284, which lacked the expression of progesterone receptor (PR) and became TNBC (Supplementary Table 1).

We then examined the antitumor activity of ribociclib in these 21 PDXs and responses to therapy were classified according to the relative change in tumor volume upon treatment (similar to the Response Evaluation Criteria In Solid Tumors (RECIST) criteria)[20,22]. We observed a complete response (CR) in one model (5%), a partial response (PR) in two (9.5%), a stable disease (SD) in two (9.5%) and a progressive disease (PD) in 18 (76%; Fig. 1B) for a total of 14% of preclinical response rate (pRR; CR + PR) and a 24% of preclinical benefit rate (pCB; CR + PR + SD). The degree of response to ribociclib was independent of the tumor growth rate of the untreated tumors, ruling out a potential bias for slower growing tumors being more sensitive. The three TNBC models were, as expected, refractory to CDK4/6 inhibition, whereas some ER+ and HER2+ PDXs responded, and others did not. We subsequently tested the sensitivity of 17 available ER+ or HER2+ PDXs to endocrine (fulvestrant or letrozole) or anti-HER2 (trastuzumab) therapies, respectively. We observed that PD was the best response in all but one case (PDX191 with SD on fulvestrant), including the 5 models sensitive to ribociclib (Fig. 1C and Fig. S1A).

Resistance to ribociclib was generated from PDX244 (ER+, PDX244LR) after prolonged drug exposure (Fig. 1D)[8] and from PDX153 (HER2+, PDX153LR), spontaneously after 7 serial passages in the absence

of drug. Both of these ribociclib-resistant models maintained the histopathological features of their respective sensitive counterpart (Supplementary Table 1). In summary, we tested the sensitivity to ribociclib in 23 BC PDXs, including two models that acquired resistance to ribociclib from the sensitive counterparts.

### PDXs expressing high p16 are resistant to ribociclib

To identify biomarkers of de novo resistance to ribociclib, we undertook genetic, transcriptomic and proteomic approaches. We firstly determined the intrinsic PAM50 subtype of the PDXs[23]. Most of the 23 PDXs showed concordant molecular and intrinsic subtypes (83%) except four PDXs expressing ER, that were categorized as basal-like (PDX313, PDX098 and STG201) or HER2-enriched (PDX225) instead of Luminal B, suggesting that they are not dependent on ER signaling. Unsurprisingly, all basal-like models (ER+ or TNBC) were resistant to ribociclib (Fig. 2A)[24]. We then performed genetic analyses of these PDXs employing a capture-based sequencing platform that detects genomic aberrations in ~410 cancer related genes (MSK-IMPACT™)[25] and analyzed whether the incidence of genetic alterations in thirteen cell cycle and PI3K-related genes correlated with ribociclib response[8–10,12,13,16,17]. We observed a trend towards *ERBB2* amplification and *CDKN2A/B* loss-of-function mutations being more frequent in ribociclib-sensitive PDXs, whereas *CCND1* amplification and loss of function mutations in *TP53* were identified amongst CDK4/6i-resistant models (Fig. 2A, S1B and Supplementary Table 3).

Comparing the mRNA expression levels of 54 cell cycle- and apoptosis-related genes in 12 ER+ or HER2+ ribociclib-resistant versus 5 ribociclib-sensitive PDXs, we found that ribociclib-resistant models expressed higher *CCNB2* ($p = 0.002$) as well as a trend towards higher *CDK1* and *CDK7* ($p < 0.1$; Fig. S1C). Moreover, the pro-apoptotic genes *BID* and *HRK* were expressed at lower levels in ribociclib-resistant PDXs compared to the sensitive ones ($p = 0.03$ and 0.02, respectively). A similar pattern of expression was observed in ribociclib-treated PDXs, with higher levels of *CCNB2* ($p = 0.03$) and *CCND1* ($p = 0.14$) in ribociclib-resistant PDXs (Fig. S1D). These results suggest that CDK4/6i-resistant tumors harbor high CDK1/cyclin B2 activity and/or undergo early adaptation to non-canonical cell cycle bypass via CDK2/cyclin D1[8].

At the protein level, both ribociclib-resistant and sensitive PDXs expressed comparable levels of ER, PR, CDK4, CDK6, cyclin D2, CDK2, and FGFR1 (Fig. S2A). Of note, none of the PDXs included in this panel harbored high-level gene copy number (CN) of *FGFR1* (Supplementary Table 3)[9,26]. Conversely, higher p16 levels ($p = 0.01$) and a trend towards low nuclear pRb levels ($p = 0.09$) were detected in resistant PDXs compared to the sensitive ones (Fig. 2B). In addition, even though the levels of cyclin E1 and cyclin D1 were similar in ribociclib-responders vs. non-responders ($p = 0.2$ and 0.4, respectively), PDXs expressing high levels of either protein were resistant to ribociclib. We further computed the accuracy of a complex biomarker composed of p16, pRb, cyclin D1 and cyclin E1 expression. This composite biomarker showed higher sensitivity (87%) and accuracy (85%) for the identification of ribociclib-resistant models compared to single or binary biomarkers (Fig. S2B). Two out of 3 ER+, basal-like PDX had high p16 levels with concomitant low pRb expression (PDX313, PDX098, Fig. 2B), which was expected given the generally reciprocal relationship between p16 and pRb (Fig. 2C)[27–29]. In addition, these two models also expressed high cyclin E1, consistent with their basal-like intrinsic subtype. In line with these observations, analysis of the ER + TCGA dataset showed the co-occurrence between low pRb and high p16 protein/mRNA expression[8] ($p = 0.003$; $p < 0.001$; Fig. S2C), and between low pRb and high cyclin E1 protein/mRNA expression ($p = 0.08$; $p < 0.001$). The latter was also observed in METABRIC. Of note, 17% of p16-high tumors in TCGA had normal levels of pRb, suggesting that high levels of p16 might indicate pRb functional loss, or be the result

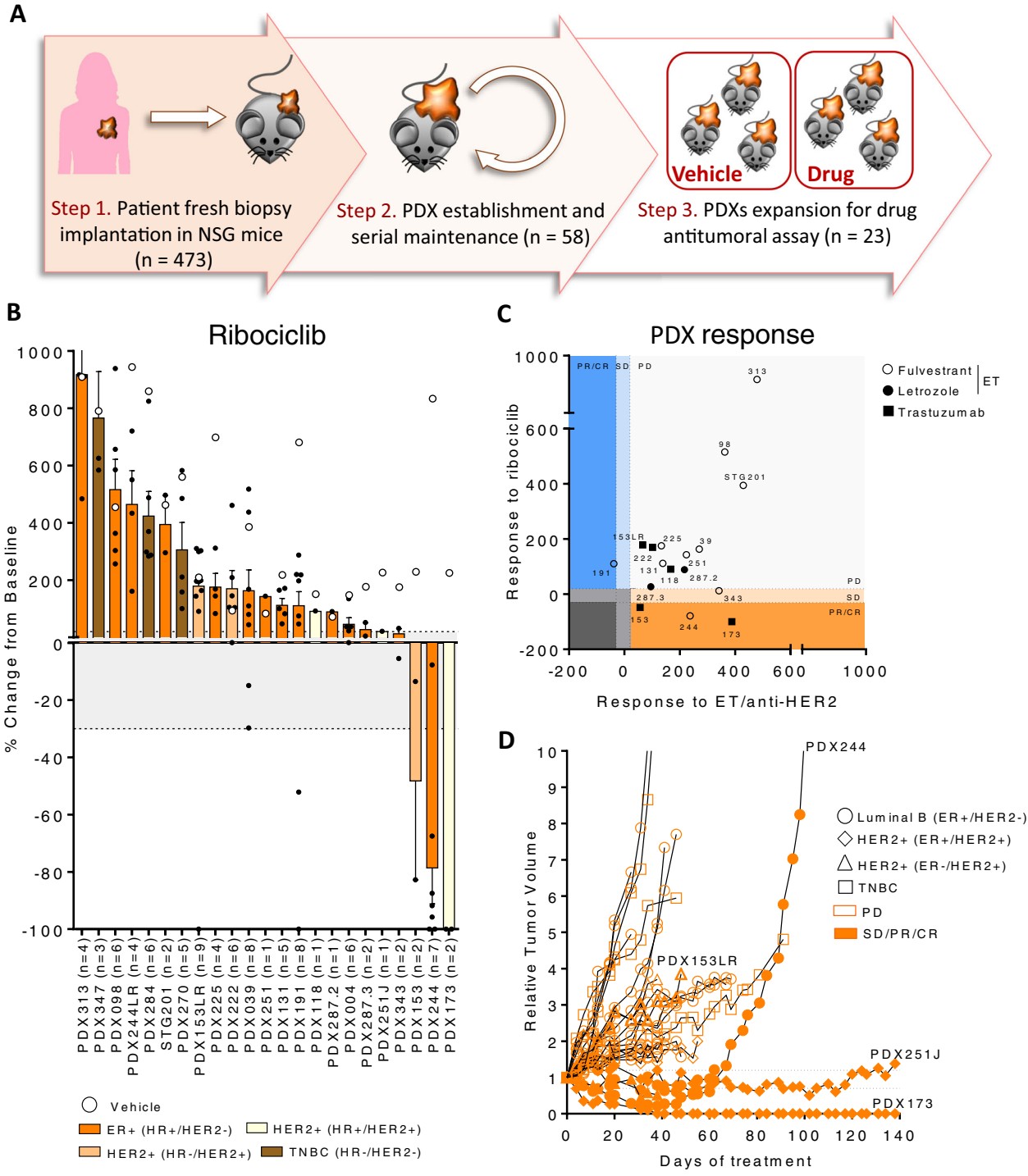

**Fig. 1 | Ribociclib monotherapy has higher antitumor activity than other targeted agents in ER+ and HER2+ BC PDXs. A** Workflow depicting the generation of BC PDX models from BC patient samples and its subsequent expansion for targeted treatment screening. **B** Waterfall plot representing the growth of $n = 23$ PDX treated with ribociclib 75 mg/kg (bars and black dots) and vehicle (white circles). The percentage change from the initial volume is shown at day 35 of treatment. Dashed lines indicate the range of PD (>20%), SD (20% to −30%) and PR/CR (<−30%). The number of tumors treated per model is indicated in brackets ($n$). Data represent mean values and error bars ± SEM. **C** Antitumor response of ribociclib (y-axis) vs.

other targeted agents (x-axis; endocrine therapy (ET) or trastuzumab) in PDXs represented as the percentage of tumor volume change compared to the initial tumor volume. Symbols represent the different targeted therapies. **D** Spaghetti plot showing the relative tumor volume change along time in 23 BC PDX treated with ribociclib 75 mg/kg. Ribociclib-sensitive models are represented with filled symbols and ribociclib-resistant with open symbols. Symbols represent the PDX's molecular subtype. Dashed lines indicate the range of PD (>1.2), SD (1.2 to −0.7) and PR/CR (<−0.7). Acquisition of ribociclib resistance in PDX244 (PDX244LR) is shown. Source data are provided as a Source Data file.

of a pRb/cyclin E1-independent oncogenic stress. In line with this, two out of 8 Luminal B PDXs resistant to ribociclib (PDX039 and PDX287.2, 25%) expressed high p16 without concomitant loss of pRb or overexpression of cyclin E1 (Fig. 2C). This data suggests that

high p16 protein expression is associated with CDK4/6i resistance regardless of the pRb status.

We further analyzed pharmacodynamic biomarkers by immuno-histochemistry as in the PreOperative-Palbociclib (POP) trial[30]. We

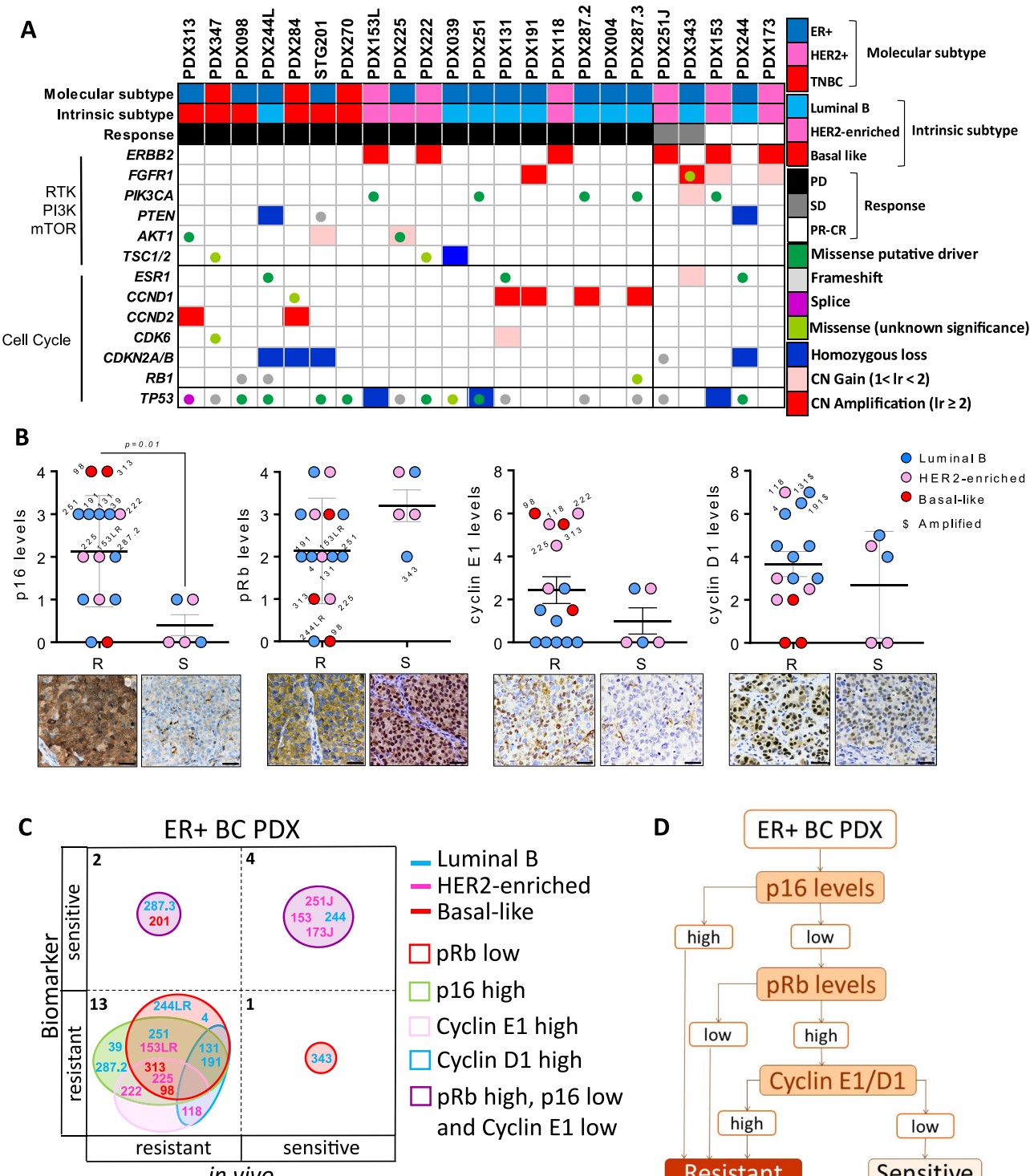

**Fig. 2 | PDXs expressing high p16 are resistant to ribociclib. A** Summary of genetic alterations in the PDX panel from Fig. 1B, including the PDX subtype classification, based on IHC (molecular subtype) or PAM50 analysis (intrinsic subtype), and the response to CDK4/6 inhibitors. Genes with similar function such as *TSC1/TSC2* or *CDKN2A/CDKN2B* were considered as one single feature. **B** Quantification of IHC staining for p16, pRb, cyclin E1 and cyclin D1 in *n* = 23 untreated PDX ribociclib-response analyzed in two independent experiments. Semiquantitative analysis was performed for pRb, p16, according to cyclin E1 and cyclin D1. Different colors indicate the PDX intrinsic subtype and $ indicates the models harboring gene amplification. Mean values ± SEM and the unpaired parametric *t*-test two-tailed *p*-value are indicated. The pictures underneath are representative bright-field images of high/low staining for each protein. Scale bar = 100 µm. R: resistant; S: sensitive. **C** Concordance analysis of the PDXs responses to ribociclib based on the analyzed biomarkers (*y*-axis) vs. the in vivo response (*x*-axis). Singleplex or multiplex biomarkers are represented by circles of different colors and the number of PDX within each category is indicated. The intrinsic subtype of each PDX is represented by different font colors. **D** Consort flow diagram for classifying the PDX responses to ribociclib based on the molecular subtype, p16, pRb and cyclin E1/D1 scores. Source data are provided as a Source Data file.

examined the percentage of KI67- and phospho-pRb (Ser807/811)-positive cells in ribociclib-sensitive and -resistant PDXs without and with ribociclib treatment and observed a decrease of KI67-positive cells as well as an unexpected increase of phospho-pRb-positive cells in ribociclib-sensitive PDXs (Fig. S2D). In summary, the basal-like intrinsic subtype classification (PAM50) and an integrated biomarker composed of p16, pRb, cyclin E1 and cyclin D1 expression levels identified models with resistance to ribociclib in PDXs (Fig. 2D).

## Biomarker validation in short-term patient-derived tumor cells (PDCs)

To validate the potential predictive biomarkers for ribociclib antitumor activity, we measured the activity of ribociclib using patient-derived cells (PDCs) grown as short-term three-dimensional (3D) ex vivo cultures on a laminin-rich extracellular matrix (Fig. 3A). PDCs were able to proliferate for at least 14 days (Fig. S2E). PDCs growth was monitored by measuring the spheroid area and the percentage of cells in the S-phase of the cell cycle using the 5-ethynyl-2′-deoxyuridine (EdU) incorporation assay. The antiproliferative activity of ribociclib was determined using sixteen representative available ex vivo cultures from the panel of 23 PDXs described above and showed that the responses to ribociclib of PDCs ex vivo agreed with that of their corresponding PDXs in vivo ($p = 0.0001$ and 0.02; Fig. 3B, Fig. S2F and S2G).

We then extended this analysis using ex vivo cultures from 14 additional ER+ BC PDXs, three of which concomitantly expressing HER2 (Supplementary Table 4). Based on p16, pRb, cyclin E1, and cyclin D1 protein expression levels, 9 of these 14 PDXs were predicted to be resistant and the remaining 5 sensitive to ribociclib (Supplementary Table 4 and Fig. 2D). Of note, 2 out of 9 models classified as ribociclib resistant due to high expression of p16 (PDX301 and PDX346) did not harbor loss of pRb. As expected, PDCs predicted as ribociclib sensitive exhibited a greater reduction in the relative spheroid area upon treatment than PDCs predicted as ribociclib-resistant ($p = 0.005$; Fig. 3C). Moreover, with the exception of PDX350 and PDX399, two models with unexpectedly high ex vivo sensitivity, the responses to ribociclib of all PDCs were in agreement with the predicted responses (Supplementary Table 4 and Fig. 3D). ROC curve analysis indicated that the change of the spheroid area upon ribociclib treatment could discriminate between sensitive and resistant PDCs with 100% sensitivity and 87.5% specificity ($p < 0.0001$; cut-off $= -25\%$; $n = 37$ models; Fig. 3E). In summary, we identified ribociclib resistant ER+ BC PDCs using a composite biomarker of p16, pRb, cyclin E1 and cyclin D1.

## p16 and cyclin D1 overexpression attenuate the response to ribociclib in ER+ BC cell lines

Previous studies have already demonstrated the impact of *RB1* loss and *CCNE1* amplification in response to CDK4/6i[8,13,16,17]. Therefore, we aimed at evaluating whether p16 and cyclin D1 overexpression can be added to the list of candidates associated with resistance to ribociclib in ER+ BC. We generated T47D and MCF7 cells (ER+) overexpressing p16 with a doxycycline-inducible system (T47D-p16, MCF7-p16). We measured the response of these cells to ribociclib, fulvestrant and the combination and found that T47D-p16 cells had 20-fold higher IC50 (half-maximal inhibitory concentration) for ribociclib and 6-fold higher for the combination of ribociclib plus fulvestrant than MOCK control cells; and MCF7-p16 cells had 17- and 24-fold IC50 values, respectively (Fig. 4A). Biochemical analysis revealed that T47D-p16 cells had higher levels of phospho-pRb (S780, $p = 0.0004$; S807/811, $p = 0.003$), cyclin E2 ($p = 0.003$) and phospho-CDK2 T160 ($p = 0.0009$) compared to control cells (Fig. 4B and Fig. S3A). Also, ribociclib treatment resulted in CDK6 upregulation in T47D cells while p16-overexpression resulted in reduced CDK6 expression in MCF7 cells, so it cannot be excluded that CDK6 modulation contributes to the CDK4/6i resistance phenotype observed in vitro. In a competition

experiment, whereby T47D-p16 cells were seeded 1:20 with control cells, p16 expression levels were upregulated after 14 days of treatment ($p = 0.0006$; Fig. 4C and Fig. S3B). Similar results were obtained with MCF7 cells, suggesting that pre-existing, low-abundant p16-overexpressing cells were positively selected upon treatment with CDK4/6i and represent a reservoir of drug-resistant cells.

To further support these findings, we posited that P18IN003[31], an inhibitor of the p18-CDK4 interaction, would also impair the binding of p16 to CDK4/6 and sensitize p16-high PDCs to CDK4/6i. Using in silico modeling, we observed that the ankyrin repeats 1, 2, 3 in p16 constitute a binding pocket that is relatively large and shallow. One part of this pocket consists largely of hydrophobic residues (Val51, Met52, Met53 and Met54) and the other part contains charged residues (Asp74, Asp84, Glu88, Arg46, and Arg87). P18IN003 fits into this pocket, with one methoxyphenyl moiety and the dihydroimidazole moiety of P18IN003 involved in H-bond interactions with Asp74 and Glu88, respectively, of p16. The other methoxyphenyl moiety of P18IN003 is solvent exposed. Comparing the predicted p16-P18IN003 interaction with the in silico model of the p16-CDK4 interaction revealed that P18IN003 would disrupt the binding of CDK4 to p16 by binding to this pocket (Fig. 4D). In ex vivo cultures, P18IN003 combined with ribociclib markedly reduced proliferation in PDC191 (p16-high, pRb-expressing model; $p = 0.0009$) but not PDC313 (p16 high, pRb low model; $p = 0.2$) compared to ribociclib monotherapy (Fig. 4E), presumably due to increased binding of ribociclib to CDK4/6[32].

The response to ribociclib, fulvestrant and their combination was also evaluated in T47D and MCF7 cells overexpressing cyclin D1 (T47D-cyclin D1 and MCF7-cyclin D1). Cyclin D1 overexpression moderately increased the IC50 values 3- to 5-fold in T47D and 2- to 3-fold in MCF7 (Fig. 4F). In line with this, T47D and MCF7 cells overexpressing cyclin D1 showed an attenuated response to downmodulation of phospho-pRb, cyclin E2 and phospho-CDK2 T160 upon treatment with ribociclib (Fig. 4G and Fig. S3C). In a competition experiment, cyclin D1 expression was upregulated after 14 days of treatment in both T47D ($p = 0.0004$) and MCF7 cell lines ($p = 0.01$; Fig. 4H and Fig. S3D). This result suggests that pre-existing, low-abundant cyclin D1-overexpressing cells were positively selected upon treatment with CDK4/6i and represent a reservoir of drug-resistant cells. In summary, we conclude that overexpression of either p16 or cyclin D1 attenuates the response of BC cells to CDK4/6i through activation of G1 checkpoint kinase activity.

## High p16 levels associated with lack of response to CDK4/6i in ER+ BC patients

Given our preclinical results, we interrogated a potential association of the aforementioned biomarkers with response to CDK4/6i in ER+ BC patients. In early-stage breast cancer, treatment with palbociclib or abemaciclib significantly reduced proliferation and the CDK4/6 downstream response, measured by KI67 and phospho-pRb (S807/811), respectively[30,33]. Here, we reanalyzed data from the abemaciclib preoperative (ABC-POP) clinical trial[33]. In total, 72 patient samples were analyzed, 33 Luminal A and 39 Luminal B. As expected, the majority of the Luminal A tumors were sensitive to CDK4/6 inhibition in terms of drop of KI67 at day 15 of treatment, a "gold standard" biomarker of endocrine sensitivity in this patient population[34]. On the contrary, we found that similar proportions of Luminal B tumors could be classified as responders (56%) and non-responders (44%; Fig. 5A). In line with results obtained in PDXs (Fig. 2B), the H-score levels of p16, but not pRb, cyclin E1 and cyclin D1, were significantly higher in resistant tumors treated with abemaciclib compared to sensitive tumors ($p = 0.008$; Fig. 5B and Fig. S4A). In the multivariable analyses, high p16 expression alone was significantly associated with a decreased probability of KI67-response ($p = 0.004$ for All tumors and $p = 0.03$ for Luminal B tumors) and we did not identify independent prognostic association of the other three biomarkers (Fig. 5C and Fig. S4B).

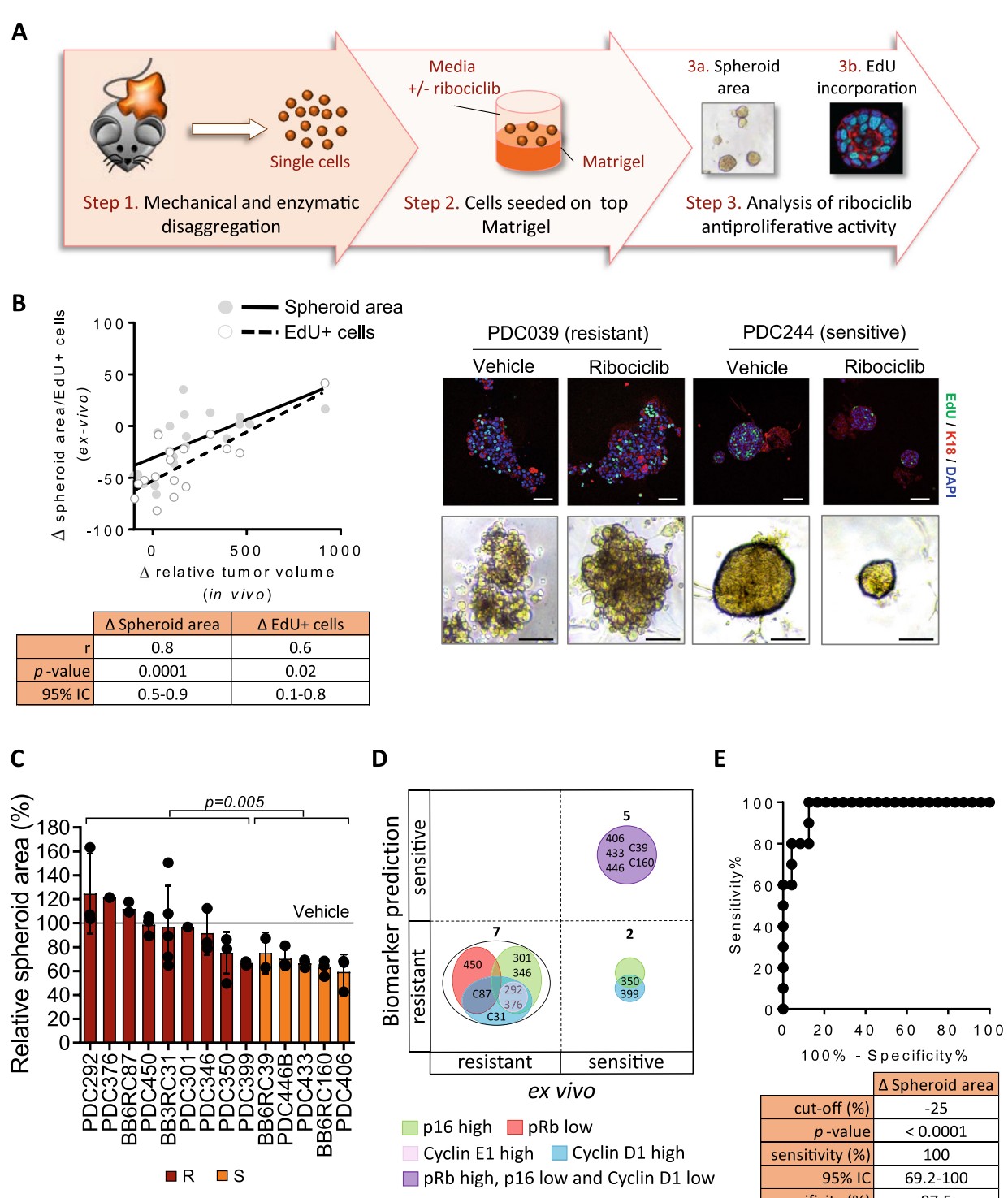

**Fig. 3 | Biomarker validation in short-term patient-derived tumor cells (PDCs).**
**A** Workflow depicting the generation of BC PDCs short-term ex vivo cultures from PDXs and the subsequent analysis of ribociclib response using two different read-outs. **B** Correlation analysis of the ex vivo response of PDCs (y-axis) vs. the in vivo response of the corresponding PDXs (x-axis), measured as change in spheroid area (open dots) or the change in EdU incorporation (filled dots) after ribociclib treatment. The Spearman's coefficient (r), two-tailed p-value and 95% of confidence interval (95% CI) for each read-out are summarized below the graph. Representative images of one ribociclib-resistant (PDC039) and one ribociclib-sensitive (PDC244) model treated with vehicle or ribociclib are shown on the right panel, namely EdU/K18 staining by confocal microscopy and organoid size by bright field microscopy. EdU is shown in green, cytokeratin 18 (K18) in red and DAPI in blue. Scale bar =

100 μm. **C** Relative spheroid area in n = 14 PDC models classified as resistant (maroon) or sensitive (orange) according to the composite biomarker after treatment with 1 μM ribociclib for 7 days. Relative data to the vehicle control (100%) is represented as mean values of three independent experiments ± SEM; unpaired parametric t-test two-tailed p-value are indicated. **D** Concordance analysis of PDXs responses to ribociclib based on biomarker prediction (y-axis) vs. the ex vivo response (x-axis). Biomarkers are represented by circles with different colors and the number of PDX within each category is indicated. **E** ROC curve of the spheroid area increment for ribociclib response prediction in n = 37 BC PDCs. Unpaired t-test two-tailed p-value (<0.0001), the sensitivity, 95% confidence interval (95% CI) and specificity are summarized below the graph. Source data are provided as a Source Data file.

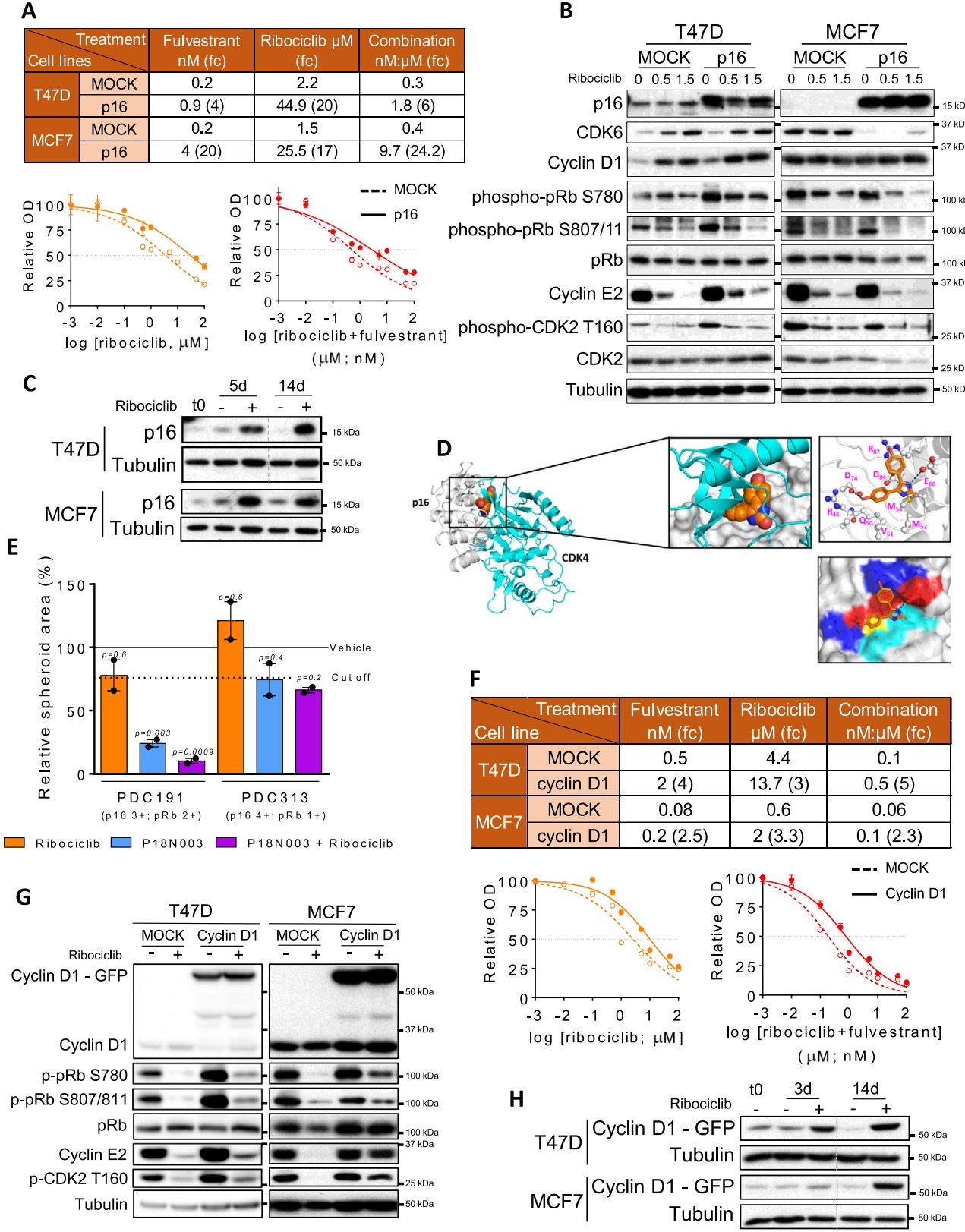

**A**

| | Treatment Cell lines | Fulvestrant nM (fc) | Ribociclib μM (fc) | Combination nM:μM (fc) |
|---|---|---|---|---|
| T47D | MOCK | 0.2 | 2.2 | 0.3 |
| | p16 | 0.9 (4) | 44.9 (20) | 1.8 (6) |
| MCF7 | MOCK | 0.2 | 1.5 | 0.4 |
| | p16 | 4 (20) | 25.5 (17) | 9.7 (24.2) |

**F**

| | Treatment Cell line | Fulvestrant nM (fc) | Ribociclib μM (fc) | Combination nM:μM (fc) |
|---|---|---|---|---|
| T47D | MOCK | 0.5 | 4.4 | 0.1 |
| | cyclin D1 | 2 (4) | 13.7 (3) | 0.5 (5) |
| MCF7 | MOCK | 0.08 | 0.6 | 0.06 |
| | cyclin D1 | 0.2 (2.5) | 2 (3.3) | 0.1 (2.3) |

Next, we tested if high p16 was also associated with lack of response to abemaciclib as single agent in the metastatic setting ($n = 17$). Higher p16 levels ($p = 0.04$) and high cyclin E1 levels ($p = 0.03$) were detected in resistant tumors ($n = 6$) compared to the sensitive ones ($n = 11$) whereas levels of pRb and cyclin D1 were not associated with CDK4/6i resistance (Fig. 5D). Interestingly, the three CDK4/6i-resistant patients with high p16 levels did not exhibit low pRb. In this cohort, multivariable analyses was not feasible due to the limited number of samples. In summary, our data shows that high protein levels of p16 are associated with resistance to CDK4/6i in both early-stage and metastatic ER+ BC tumors.

**Fig. 4 | p16 and cyclin D1 overexpression attenuate the response to ribociclib in ER⁺ BC cell lines. A** Half-maximal inhibitory concentration (IC₅₀) and IC₅₀ fold-change (fc) values for ribociclib, fulvestrant and the combination of drugs in T47D and MCF7 cells overexpressing p16 compared to control cells (MOCK), treated for 6 days and evaluated as shown underneath. Data are presented as mean values ± SEM of at least three independent experiments. **B** Immunoblot of the indicated proteins in MOCK and p16 overexpressing T47D and MCF7 cells untreated or treated with ribociclib for 5 days at the indicated concentrations. **C** Immunoblot of the indicated proteins in a cell competition assay. Cells were treated with vehicle or ribociclib 1 μM for the indicated days. **D** Comparison of structural models for the complexes of p16 bound to P18IN003 and p16 bound to CDK4. Gray cartoon is p16, orange spheres represent P18IN003, cyan cartoon is CDK4; zoomed view are the binding pocket with p16 shown as gray surface and highlighting the residues in the binding pocket of p16 as sticks and the

hydrogen bonds made between P18IN003 and p16 shown as black dashed lines. **E** Relative spheroid area of PDC191 and PDC313 after treatment with 1 μM ribociclib, 20 nM P18IN003 and the combination for 7 days. Data are presented as mean values of three independent experiments ± SEM. Two-tailed p-values are based on the one-way ANOVA test with Tukey's method correction compared with the vehicle (black line). Dashed line indicates the optimal cut-off established in Fig. 3E. p16 and pRb scores of each PDC are indicated. **F** IC₅₀ and IC₅₀ fc values for ribociclib, fulvestrant and the combination of drugs in T47D and MCF7 cells overexpressing cyclin D1, compared to control cells (MOCK), treated for 6 days and evaluated as shown underneath. Data are presented as mean values ± SEM from at least three independent experiments. **G** Immunoblot of indicated proteins in control (MOCK) and cyclin D1-overexpressing T47D and MCF7 cells untreated or treated with 0.5 μM ribociclib for 48 h. **H** Immunoblot as indicated for panel (**C**). Source data are provided as a Source Data file.

## Acquisition of subclonal RB1 mutations as mechanism of acquired resistance to ribociclib in tumors with RB1 heterozygous loss

We next investigated mechanisms of acquired resistance in PDXs and posited that tumors with an underlying *RB1* heterozygous loss tend to acquire *RB1* point mutations that result in CDK4/6i resistance. This hypotesis was based on the evidence from one patient with matched pre/post-CDK4/6i tumor sample and the relatively low frequency of acquired *RB1* point mutations after treatment with palbociclib[15,16,35]. We generated eight derivatives from PDX244 that became refractory to ribociclib treatment overtime[8]. Sequencing data from the sensitive tumor (2R) revealed a *CDKN2A* homozygous loss and concomitant *RB1* heterozygous loss (Fig. 6A). Protein analysis confirmed the lack of p16 and normal pRb levels in this model (Fig. 6A). Three out of 8 ribociclib resistant tumors (16L, 16R, and 18R) acquired deleterious mutations in *RB1* (p.M695Nfs*26, p.K810* and p.X180_splice, respectively) and tumors 19L and 19R underwent a further reduction in the *RB1* copy number (from −0.9 to −1.5 and −2.0, respectively), suggesting the acquisition of homozygous *RB1* loss. Protein analysis by IHC confirmed the total or subclonal loss of pRb expression in these tumors (16L and 19L vs. 16R and 18R). Intriguingly, a sixth tumor (15R) also showed partial loss of pRb expression without a detectable underlying genetic alteration in *RB1*, suggesting alternative mechanisms that regulate *RB1* gene expression. Regarding 15L and 17L, we observed an increment in *CDKN2A* CN log ratio (from −4.2 to −0.4) along with the restoration of p16 expression (Fig. 6A), suggesting that selection of tumor cells retaining normal *CDKN2A* CN was favored upon treatment. Cyclin E1 and cyclin D1 did not show any alteration at the gene or protein levels in any of the acquired-resistant tumors (Fig. 6A).

In addition to the aforementioned model, we developed paired PDXs from a patient who received palbociclib plus letrozole before treatment initiation (PDX476.1) and after 12 cycles of treatment at the time of disease progression (PDX476.2). Similar to the respective patient tumors, PDC476.1 was sensitive to palbociclib whereas PDC476.2 was resistant (Fig. S4C). Genetic analysis showed that PDX476.1 harbored a heterozygous *RB1* loss (CN = 0.9) but still expressed pRb. In contrast, PDX476.2 lost pRb protein expression, suggesting that it may be the mechanism responsible of tumor progression (Fig. 6B).

Next, we interrogated the prognostic implications of *RB1* heterozygous loss in ER⁺ BC patients. In two out of the three cohorts analyzed, patients with tumors harboring *RB1* heterozygous loss showed significantly poorer clinical outcome compared to patients with unaltered *RB1* tumors in terms of disease-free survival (DFS), overall survival (OS) or days of treatment (DOT; Fig. 6C). In order to analyze the role of *RB1* heterozygous loss as biomarker of resistance to CDK4/6i in patients, we obtained genomic and clinical data of metastatic BC patients included in the Hartwig Medical Foundation (HMF) cohort[36]. Out of 582 patients diagnosed with metastatic BC, 71 received CDK4/6i. To test whether concomitant heterozygous deletion and

mutation of the *RB1* gene appears preferentially amongst patients who had received CDK4/6 inhibitors, we applied a multivariable logistic regression. We found a significant association between the double hit (mutation and heterozygous deletion) and the prior exposure of the patient to CDK4/6i as part of their treatment (p = 0.003, Fig. 6D). No significant association was found between any alteration (only mutation, only deletion or both) affecting *RB1* and the previous exposure to CDK4/6 inhibitors (Fig. S4D), implying that only patients with a double hit in *RB1* have an increased likelihood of having received CDK4/6i. Altogether, these data suggest that tumors harboring heterozygous *RB1* loss are susceptible of acquiring a second hit in the *RB1* gene and becoming resistant to CDK4/6i.

## The PI3K inhibitor alpelisib sensitizes non-basal-like BC PDX to ribociclib

Preclinical studies[37] and a phase III clinical study (SOLAR-1, NCT02437318[18]) have shown that PI3K-α inhibitors, such as alpelisib, are able to sensitize *PIK3CA*-mutant tumors to ET. Moreover, preliminary results from the BYLieve clinical trial have shown that the proportion of patients treated with alpelisib plus fulvestrant without disease progression at 6 months (CBR) was 50.4%, showing clear efficacy of PI3K-α inhibitors post-CDK4/6i. Moreover, this benefit is maintained regardless of the duration of response to the previous CDK4/6i-based treatment or presence of CDK4/6i resistance genes in circulating tumor DNA (NCT03056755[38–40]). Here, we interrogated whether alpelisib can sensitize ribociclib-resistant PDXs to CDK4/6i.

In our PDX panel, alpelisib monotherapy resulted in a pCB of 43% compared to 25% of ribociclib alone (Fig. 7A). All HER2⁺ PDXs tested were sensitive to alpelisib, including the ribociclib-resistant PDX153LR, PDX222, and PDX118. All the *PIK3CA* mutated tumors were also sensitive to alpelisib except PDX287.3, which was derived from a patient's tumor that progressed while being treated with the PI3Ki GDC-0032 plus letrozole (Fig. S4E). As expected, all 3 TNBC PDXs were resistant to alpelisib since none of them harbored a *PIK3CA* mutation.

Combined treatment of ribociclib and alpelisib resulted in a pCB of 78% (Fig. 7A). We noticed that all basal-like PDXs by PAM50 exhibited PD or SD as best response. Similarly, combination of palbociclib plus GDC-0032 also showed improved antitumor activity compared to either one alone in PDX287.2 and PDX287.3 that derived from a patient's tumors collected on-treatment or after progression with GDC-0032 plus letrozole, respectively (Fig. S4F). Analysis of pharmacodynamic biomarkers showed that both KI67 and phospho-pRb S807/811 were downregulated in PDXs that responded to the drug combination (p < 0.001 and p < 0.01, respectively), but not in the resistant ones (Fig. 7B). Because of concerns regarding the safety of the combination of alpelisib plus ribociclib, we conducted de-escalation experiments showing that dose reduction of either ribociclib or alpelisib resulted in similar antitumor activity as the full dose tested in both PDX039 and PDX191. Dose reduction of both drugs, however, resulted in an attenuated efficacy (Fig. 7C). Of note, both

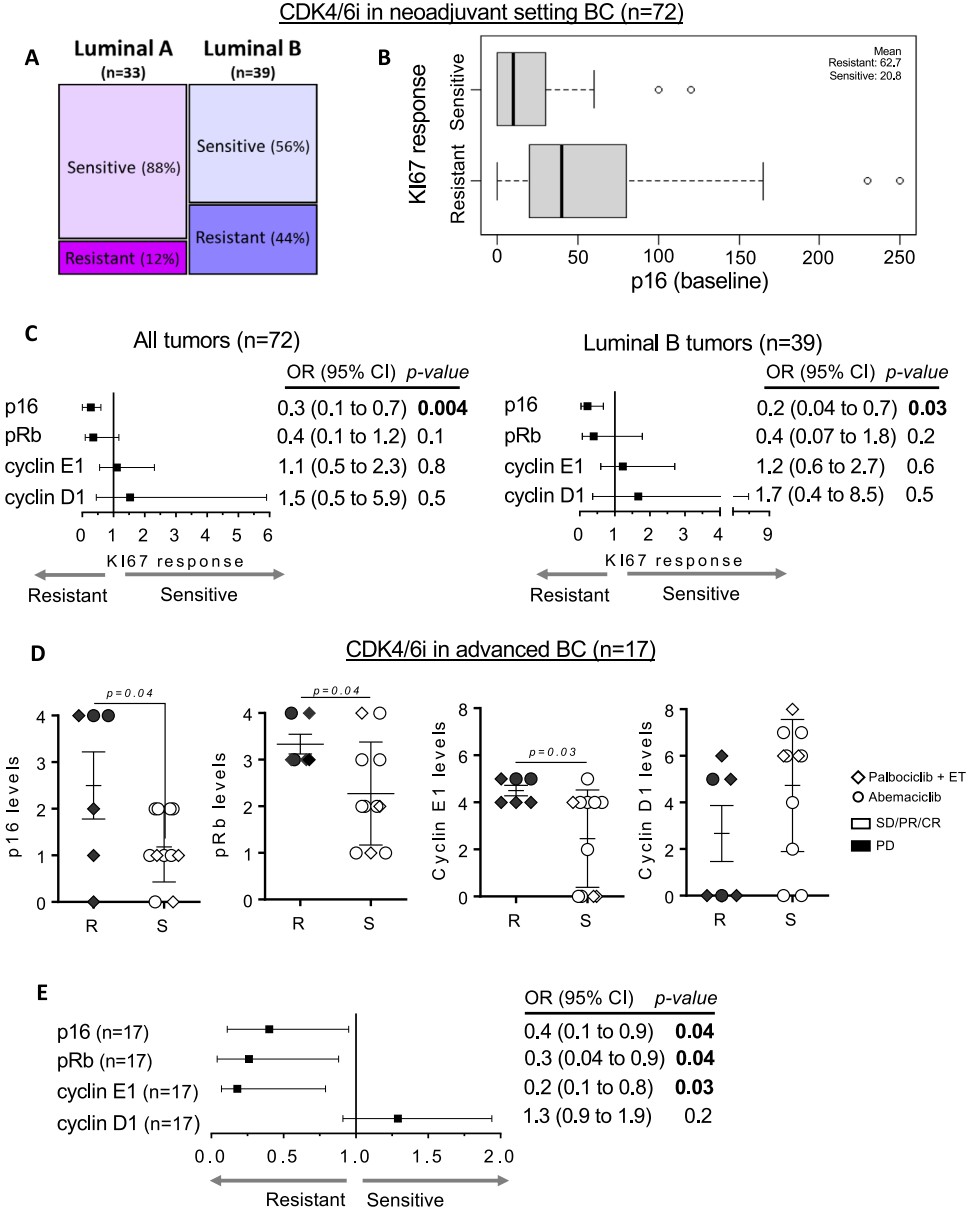

**Fig. 5 | High p16 levels associated with lack of response to CDK4/6i in ER⁺ BC patients. A** Percentage distribution of sensitivity to abemaciclib after 15 days of treatment in the neoadjuvant setting in the ABC-POP trial (*n* = 72). Tumors were classified as Luminal A if %KI67 < 15 or as Luminal B if %KI67 ≥ 15. Tumors showing ln KI67 < 1 at day 15 were considered sensitive and those with ln KI67 ≥ 1 were resistant to the studied drug. **B** Box and whiskers plot showing a logistic model to evaluate the effect of p16 on the response to abemaciclib (*n* = 39 Luminal B tumors). Box represents the median and the 25th and 75th percentiles, whiskers show the largest and smallest value. The mean value of each subgroup is indicated. **C** Forest plots displaying the odds ratios (OR; black squares) ±95% confidence intervals (CI; horizontal lines) of the association between the indicated biomarkers and the percentage of KI67 after treatment

with abemaciclib for all tumors (*n* = 72; left panel) or Luminal B tumors (*n* = 39; right panel). Two-sided *p*-values from Wald tests in logistic models are provided. **D** Quantification of p16, pRb, cyclin E1 and cyclin D1 in a cohort of *n* = 10 advanced BC tumors detected by IHC semiquantitatively (pRb p16 cyclin E1 and cyclin D1) displayed according to the patient's response to abemaciclib. Symbols indicate different CDK4/6i treatments. Mean values ± SEM and unpaired parametric *t*-test two-tailed *p*-value are indicated R: resistant; S: sensitive. **E** Forest plot displaying the odds ratios (OR; black squares) ± 95% confidence intervals (CI; horizontal lines) of the association between the indicated biomarkers and the patient's response to the study treatments (*n* = 10 tumors). Fisher's exact test two-tailed *p*-values are indicated. Source data are provided as a Source Data file.

dose reductions were within the range of doses being tested/recommended in the clinic for these combinations[41].

Importantly, we demonstrated that ribociclib plus alpelisib (or the triple combination with fulvestrant) was effective in two *PIK3CA*-wt, *ESR1*-mut PDXs with primary resistance to ribociclib plus fulvestrant (PDX131 and PDX244LR#18 R, Fig. S4G and Fig. 7D) and in a PDX from a patient who showed an early progression when treated with palbociclib plus letrozole (PDX450, *PIK3CA* and *ESR1* mutant; Fig. 7E). Similar results were obtained ex vivo with a *PIK3CA*-wt PDX from a patient who received

palbociclib plus letrozole: treatment of PDC476.2 with palbociclib plus alpelisib resulted in higher reduction of the spheroid area than palbociclib plus fulvestrant (Fig. S4C). The triple combination of ribociclib plus alpelisib and fulvestrant was the most efficacious controlling proliferation and, biochemically, increasing the levels of PARP1 cleavage (Fig. 7F). Altogether, these results suggest that the combination of a PI3Ki with a CDK4/6i, with or without ET, is a valid therapeutic option for the treatment of ER⁺ BC tumors after progression on CDK4/6i plus ET, independently of *PIK3CA*, *ESR1* mutation status or pRb expression.

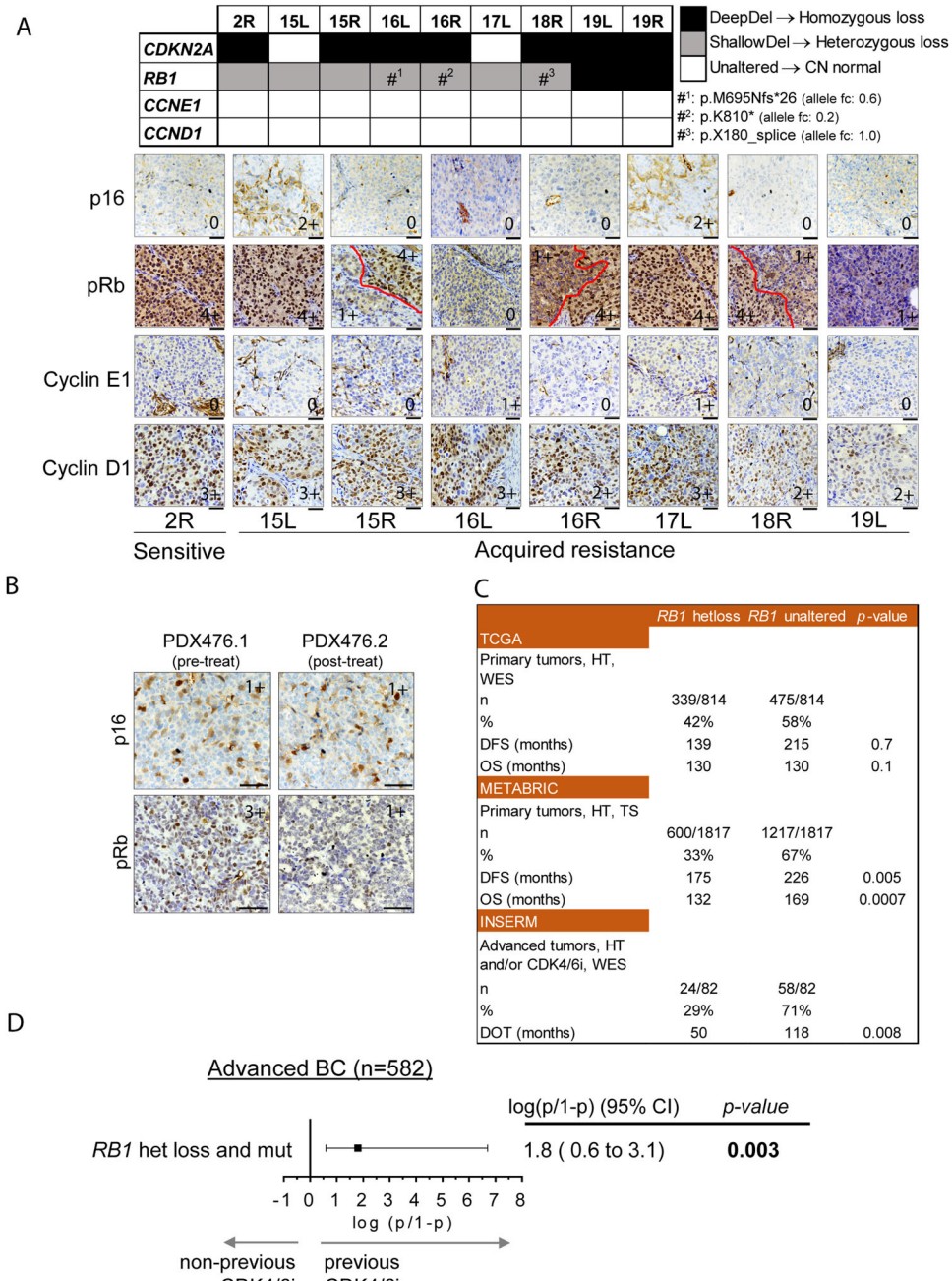

**Fig. 6 | Acquisition of subclonal *RB1* mutations as mechanism of acquired resistance to ribociclib in tumors with *RB1* heterozygous loss. A** On the top, copy number and mutation status of *CDKN2A*, *RB1*, *CCNE1* and *CCND1* in tumors derived from PDX244, including an untreated sensitive tumor (2R) or tumors with acquired resistance to ribociclib (15L to 19R). Deep-Del (homozygous loss), CN < −1; Shallow-Del (heterozygous loss), −1 ≤ CN < −0.4; Unaltered, CN ≥ −0.4. Hashtags indicate tumors that acquired deleterious mutations in *RB1*. On the bottom, representative pictures showing IHC staining of p16, pRb, cyclin E1 and cyclin D1 from the indicated tumors (bottom). Dashed-red lines mark off areas with different protein staining intensity and protein score is indicated. For 19R there was no FFPE tumor available. Scale bar = 100 μm. **B** IHC staining of p16 and pRb in representative FFPE sections from PDX476.1 and PDX476.2. Protein score is provided. Scale bar = 100 μm. **C** Clinical outcome expressed in months for the indicated clinical endpoint in patients with ER⁺/HER2⁻, *RB1* unaltered tumors vs. those harboring tumors with *RB1* heterozygous loss. Data and statistical analysis were extracted from the cBioportal. HT hormone therapy, WES whole-exome sequencing, TS targeted-sequencing, n number of patients, DFS disease-free survival, OS overall survival, DOT days of treatment. **D** Association between *RB1* double hit alterations (concomitant mutation and deletion) and prior exposure to CDK4/6 inhibitors in metastatic BC patients (*n* = 582 tumors). Black square represent the logit value. Multivariable logistic regression two-tailed unadjusted *p*-value and the 95% confidence intervals (CI; horizontal segment represents) for the test are shown. Source data are provided as a Source Data file.

## Discussion

15–30% of ER⁺ metastatic BC progress rapidly when treated with CDK4/6i plus ET. In general, the subsequent line of therapy has a short duration, lasting less than 6 months[42,43]. Therefore, it is important to identify the group of patients that are not likely to benefit from CDK4/6i plus ET to avoid unnecessary toxicity, to reduce unnecessary costs and to provide more effective alternative treatments. Targeting mTORC1 with everolimus or PI3K with alpelisib in combination with aromatase inhibitors (AI) or fulvestrant, respectively, are currently available therapies but have some limitations. Everolimus approval for

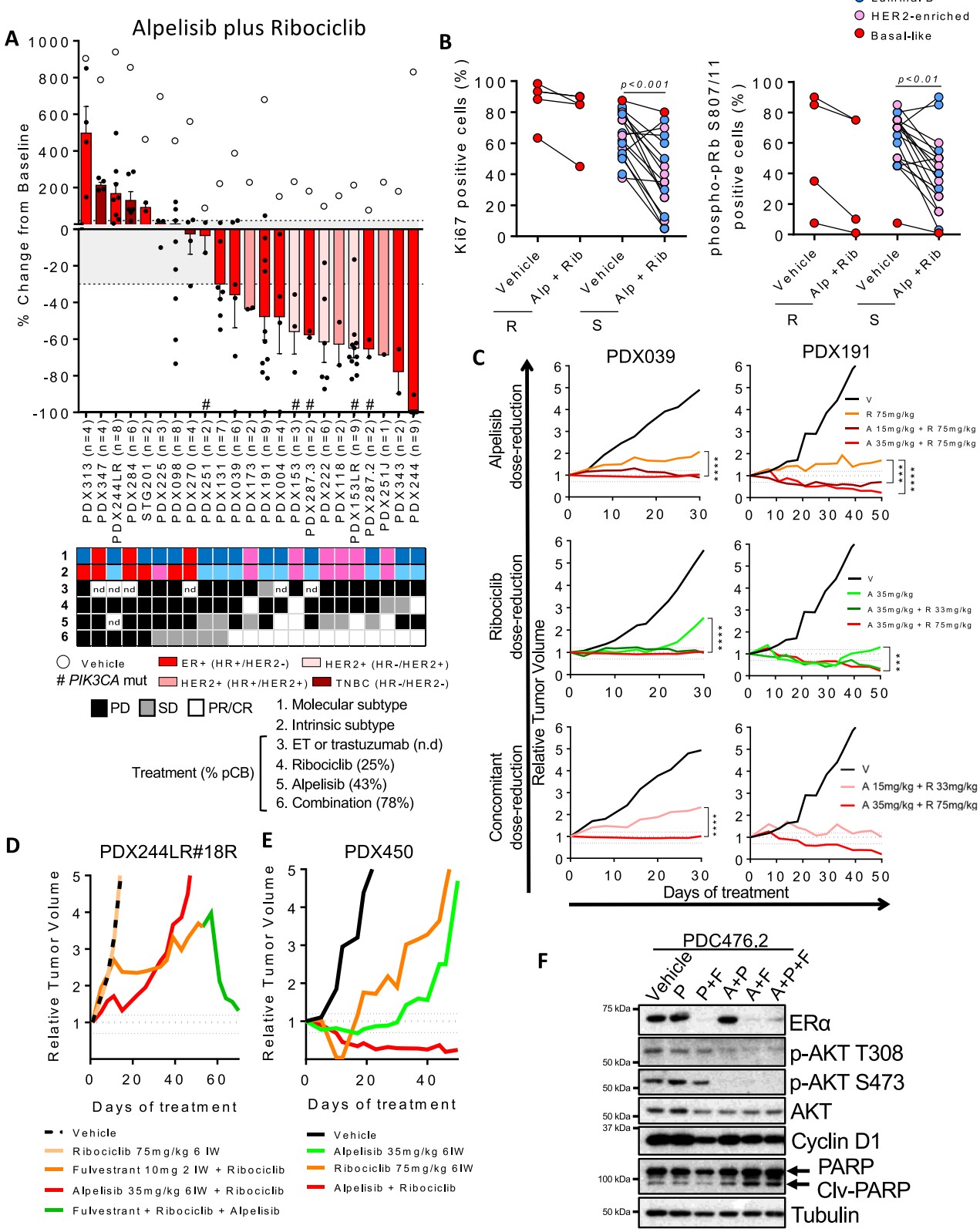

metastatic ER⁺ BC is in combination with an AI, but AI is not active in patients with *ESR1*-mutant tumors[44]. On the other hand, alpelisib is only recommended for patients with BC harboring *PIK3CA* mutations[18]. In this work, we explored biomarkers of resistance to CDK4/6i and the efficacy of a therapeutic strategy based on CDK4/6i plus PI3Ki. We identified that overexpression of p16 or cyclin D1 is associated with an impaired response to CDK4/6i and that a CDK4/6i in combination with

a PI3Ki is effective in CDK4/6i-resistant ER⁺, non-basal-like PDXs, independently of the *PI3KCA* or *ESR1* mutation status. Moreover, triple combinations of a CDK4/6i with a PI3Ki and ET are active in PDXs harboring biallelic *RB1* mutation (or pRb protein loss).

Although CDK4/6i are mostly administrated in combination with ET, abemaciclib has shown activity in ER⁺/HER2-negative metastatic BC that progressed on ET and chemotherapy[4]. Also, palbociclib has shown

**Fig. 7 | PI3K inhibition sensitizes non-basal-like BC PDX to CDK4/6i. A** Waterfall plot showing the growth of *n* = 23 PDX treated with ribociclib 75 mg/kg plus alpelisib 35 mg/kg (bars and black dots) and vehicle (white circles). The percentage change from the initial volume is shown at day 35 of treatment. Dashed lines indicate the range of PD (>20%), SD (20% to −30%) and PR/CR (<−30%). The number of tumors treated per model is indicated in the brackets (n). Hashtags (#) indicate models harboring mutations in *PIK3CA*. Data represent mean values and error bars ± SEM. Boxes underneath show the molecular and intrinsic tumor subtype as well as their responses to the indicated treatments. The preclinical benefit to each drug is indicated as percentage in brackets. n.d, not determined. **B** IHC analysis of KI67 (left graph; *p*-value < 0.001) and phospho-pRb S807/811 (right graph; *p*-value < 0.01) in vehicle and alpelisib plus ribociclib-treated PDXs according to the PDX response to alpelisib plus ribociclib. Data are presented as mean values of each PDX. Two-tailed

*p*-values are based on Mann-Whitney U test for all biological replicates. Different colors represent the tumor's intrinsic subtype. R, resistant; S, sensitive; Alp, alpelisib; Rib, ribociclib. **C** Relative tumor growth of PDX039 and PDX191 after treatment with the indicated drugs and time. Dashed lines indicate the range of PD (>1.2), SD (1.2 to −0.7) and PR/CR (<−0.7). Two-tailed *p*-values are based on the two-way ANOVA test with Bonferroni's method correction. V, vehicle; R, ribociclib; A, alpelisib. **D, E** Relative tumor growth of the ribociclib-resistant PDX244LR#18R and PDX450 after treatment with the indicated drug(s) for the indicated period of time, respectively. Dashed lines indicate the range of PD (>1.2), SD (1.2 to −0.7) and PR/CR (<−0.7). **F** Immunoblot of the indicated proteins in PDC476.2 treated with vehicle or 500 nM palbociclib as single-agent or combined with 100 nM fulvestrant and/or 2.5 μM alpelisib in ex vivo cultures for 48 h. At least two independent experiments were conducted. Source data are provided as a Source Data file.

a signal of clinical activity in post-menopausal women with advanced ER+/HER2-negative BC having received prior ET[45]. This data with abemaciclib and palbociclib monotherapy support the examination of CDK4/6i monotherapy-specific mechanisms of sensitivity. We observed that *CDKN2A/B* were more frequently disrupted in ribociclib-sensitive models and the best PDX responder lacked expression of p16. Similarly, in the Phase I trial of abemaciclib the best BC responder harbored a concomitant deletion of *CDKN2A* and *CDKN2B* in her tumor[46]. This observation is consistent with the increased sensitivity to palbociclib observed in 544 cancer cell lines in association with *CDKN2A* inactivation by DNA methylation and with the limited capacity of current CDK4/6i to inhibit its target when it is bound to INK4 proteins or to p27[Kip1][14,32,47].

*FGFR1* gain and *TP53* mutation detected in circulating tumor DNA have recently emerged as markers of early progression on CDK4/6i treatment, albeit they are also associated with poor prognosis irrespective of palbociclib treatment[9,26]. Although *CCND1* and *CDKN2A* individually have not been associated with lack of response to CDK4/6i plus ET, results from the BioItaLee (NCT03439046) and PALOMA-1 (NCT00721409) trials suggest that genes in the CDK4/6 pathway are also biomarkers of de novo resistance[2,48]. High p16 protein has been previously described as a surrogate biomarker of pRb loss in several tumor types[28,49]. Here, we found similar results analyzing 814 ER+ invasive breast carcinomas from the TCGA dataset. However, we also observed that 17% of the p16-high tumors from TCGA do not express low pRb, suggesting that p16 might independently predict CDK4/6i-resistance. This phenotype might have been relevant in CDK4/6i-resistant tumors from the ABC-POP trial[33] in which p16, but not pRb, was associated with resistance to abemaciclib. The identification of p16 overexpression as biomarker of resistance to CDK4/6i is in contrast to the results observed in PALOMA-2 and MONALEESA-2 and could be due to the specific cut-offs applied or the specimens utilized[50,51]. However, our data is in line with recent obsevations showing that INK4 proteins such as p16 and p18 induce resistance in CDK6-overexpressing tumors[14]. Specifically, INK4 proteins preferentially bind CDK6 over CDK4 and distort the ATP/drug binding pocket of CDK6 in a manner that binding of ATP is favored over palbociclib, inducing drug resistance. These findings are relevant for the clinic because multiple genetic alterations lead to CDK6-mediated resistance (e.g. *FAT1, PTEN*). A PROTAC drug to specifically degrade CDK6 was able to overcome CDK4/6i resistance. Our results further show that targeting the p16-CDK4/6 interaction sensitizes p16-overexpressing tumor cells to CDK4/6i.

Acquired or pre-existing *RB1* mutations has also been associated with resistance to ET plus CDK4/6i[13,16,26]. Wander et al. described one potential mechanism being the biallelic disruption in *RB1* in a patient who progressed to ET plus palbociclib, and harbored an *RB1* heterozygous loss in her pre-treatment tumor sample[35]. This evidence is in line with our results showing that most of the tumors from the ribociclib-sensitive PDX244, that harbors an *RB1* heterozygous loss, became resistant to CDK4/6

blockade due to acquisition of a second hit in *RB1*. In patients, we observed that *RB1* heterozygous loss is associated with worse prognosis and that a double hit in *RB1* is more likely to occur in patients who received a CDK4/6i than in those with ET alone. At least in part, this might be explained by the haploinsufficiency of pRb in its contribution to DNA repair in the S-phase checkpoint and initiation of DNA replication; cancer cells with only one copy of *RB1* exhibit a genomic instability phenotype[52]. Therefore, tumors harboring *RB1* heterozygous loss might be prone to acquire a second hit in *RB1* and resistance to CDK4/6 blockade. As we have shown, tumors with *RB1* loss are sensitive to the triple combination of CDK4/6 plus PI3Ki and ET or might be sensitive to other targeted treatments such as inhibitors of the Aurora kinases AURKA and AURKB[53].

In closing, therapies for metastatic BC that progress on or after CDK4/6i treatment currently being tested in the clinic include: (1) ET or more potent SERDs in endocrine sensitive patients, albeit this patient population is difficult to identify (NCT02338349, NCT03616587); (2) continuation of CDK4/6i with a different ET backbone (NCT03616587; NCT03809988); (3) continuation of ET with a different CDK4/6i; for example with abemaciclib, because it has additional targets including CDK1/2 complexes[54]; (4) different ET combined with a PI3Ki in *PI3KCA* mutant ER+ BC[18,38–40]; or (5) different ET combined with an AKT inhibitor in *PIK3CA/AKT1/PTEN* altered tumors (NCT04305496). In this context, our previously reported preclinical data suggests that *PTEN* alterations alone do not result in benefit to AKT inhibitors[55]. Here, we report that the combination of ribociclib with the PI3K inhibitor alpelisib (or palbociclib with GDC-0032) has antitumor activity in non-basal-like PDXs independently of *PIK3CA* or *ESR1* mutation, extending the evidence from cell line-derived xenoimplant models[11,56,57]. Of note, PAM50-based intrinsic subtyping has become a potential indicator of benefit to CDK4/6i and our data highlights the continued translational relevance of PAM50 which is not currently used in standard practice for early-stage or metastatic breast cancer[58–60]. In addition, we demonstrated antitumor efficacy of the triple combination (CDK4/6i plus PI3Ki and ET) in PDXs harboring *RB1* mutation, which may be an appropriate first-line treatment strategy for patients harboring heterozygous *RB1* loss. Finally, we provide rationale for de-escalation strategies, either by omission of ET or reduction of the PI3Ki/CDK4/6i dose.

Altogether, this study identifies high p16 protein levels and heterozygous *RB1* loss as biomarkers of resistance to CDK4/6i treatment and suggests that CDK4/6i plus PI3Ki may be effective in non-basal-like tumors that progress to CDK4/6i and ET, independently of the *PIK3CA*, *ESR1* or *RB1* status.

## Methods
### Study design
This study was designed to identify predictive biomarkers of response to ribociclib that can be effectively used for patient stratification. We assessed ribociclib activity in a cohort of 37 patient-derived xenograft

models from primary/metastatic breast cancer patients. All animal procedures were approved by the Ethics Committee of Animal Research of the Vall d´Hebron Institute of Oncology and by the Catalan Government (FUE-2020-01541918) and were conformed to the principles of the WMA Declaration of Helsinki, the Department of Health and Human Services Belmont Report and following the European Union's animal care directive (2010/63/EU). Use of PDXs from other laboratories was approved by the National Research Ethics Service, Cambridgeshire 2 REC (RED reference number: 08/H0308/178 and http://caldaslab.crik.cam.ac.uk/bcape/) or by the Central Office for Research Ethics Committee study number 05/Q1402/25. For ethical issues, in vivo experiments were ended when the total tumor volume of a mouse surpassed the maximal tumor size permitted by the ethics committee, namely 1500 mm³, or a decline in mouse welfare was observed. Tumors were harvested and formaldehyde and flash-frozen for posterior proteomic and genomic analyses.

For PDX generation, we obtained fresh tumor samples from the Vall d´Hebron University Hospital and following the institutional guidelines. Informed written patient consent, approved by the Ethics Committee for Clinical Research and Animal Research of Vall d'Hebron Hospital (PR(AG)130/2015), was obtained for the use of these patient samples.

To study the predictive value of cell cycle biomarkers, 72 patients of the 105 enrolled in the ABC-POP trial (NCT02831530) were subject to analysis of pRb, p16, cyclin E1 and cyclin D1 in their tumors. This analysis was blinded to the study endpoint, namely antiproliferative response defined as natural logarithm of Ki67 expression at day 15 < 1. Briefly, untreated females aged 18 years or older who were diagnosed with HR-positive, non-metastatic invasive breast carcinoma and signed an informed consent were selected. Patients received abemaciclib (150 mg twice daily for 14 days) or no treatment before surgery. FFPE tumor samples were obtained at baseline and upon surgery for biomarker analysis. Regarding the metastatic cohort, FFPE material representative of the disease was obtained from HR-positive metastatic breast cancer patients aged 18 years or older who signed an IRB-approved informed consent form and received treatment at the Vall d'Hebron University Hospital with a CDK4/6 inhibitor as monotherapy or in combination with HT. Patients were considered sensitive to the treatment in the metastatic setting if a clinical benefit (defined by RECIST criteria) was achieved and maintained for a period ≥ 6 months and/or ≥ 10 cycles of treatment. pRb, p16, cyclin E1 and cyclin D1 were analyzed in their tumors.

### Antibodies and reagents

Primary antibodies used for immunohistochemistry (IHC) were cyclin D1 (RM9104, 1:100) from ThermoScientific; pRb (554136, 1:100) from BD Pharmigen; phospho-pRb S807/811 (8516, 1:300) from Cell Signaling Technology; cyclin E1 (05-363, 1:300) from Millipore; p16 (725-4713), ER (790-4324), PR (790-2223), Ki67 (790-4286) and HER2 (790-2991) from Ventana Medical Systems, Roche. Primary antibodies used for immunofluorescence (IF) were Cytokeratin 18 (ab133263; 1:500) and Alexa Fluor® 568 anti-Vimentin (ab202504; 1:500) both from Abcam and secondary antibody was Alexa Fluor® 488 goat anti-rabbit IgG (A48282, 1:2000). Primary antibodies used for Western blot were CDK4 (12790), cyclin D2 (3741), phospho-pRb S807/811 (9308), phospho-pRb S780 (9307), phospho-CDK2 T160 (2561), phospho-AKT T308 (2965), phospho-AKT S473 (9271), AKT (9272), PARP (9542) and FGFR1 (3472) from Cell Signaling Technology; CDK6 (ab124821), cyclin D1 (ab40754) and cyclin E2 (ab40890) from Abcam; Tubulin (T-9026) from Sigma; cyclin E1 (sc-481), CDK2 (sc-163) and, human GAPDH (sc137179) from Santa Cruz Biotechnology; p16-INKA (10883-I-AP) from ProteinTech; pRb (554136) from BD Pharmigen; ER-alpha (MS-315-P0) from Neomarkers. All primary antibodies were diluted 1:1000 except for human GAPDH and Tubulin that were diluted 1:5000. Secondary antibodies used for Western blot were goat anti-rabbit IgG HRP linked whole antibody (NA934) and goat anti-mouse IhG HRP linked

whole antibody (NA931) from Sigma-Aldrich. All secondary antibodies were diluted 1:2000. Ribociclib (LEE11) and alpelisib (BYL719) were provided by Novartis. Commercial trastuzumab (Herceptin) was obtained from a pharmacy. Fulvestrant and letrozole were purchased from Selleckchem. P18IN003 was purchased from Glixxlabs. Lenti ORF clone of human cyclin D1 (*CCND1*) mGFP tagged (RC204957L2) and Lenti-C-mGFP tagged empty vector (PS100071) were purchased from Origene. pLX401-INK4A vector was a gift from William Hahn (Addgene plasmid#121919; http://n2t.net/addgene:121919;RRID: Addgene 121919).

### Cell lines

293T (CRL-3216), MCF7 (HTB-22) and T47D (HTB-133) cell lines were obtained from ATCC and maintained according to the manufacturer's instructions. Cell lines were authenticated utilizing short tandem repeat (STR) profiling (FTA Sample Collection Kit for Human Cell Authentication Service; ATCC services). The submitted profile was an exact match for the ATCC human cell lines in the ATCC STR database. Mycoplasma test was performed every 5 passages (MycoAlert™ Mycoplasma Detection Kit; LONZA). Both MCF7 and T47D cell lines were grown in RPMI 1640 with GlutaMAX medium (Gibco) supplemented with 10% of heat inactivated fetal bovine serum (Gibco) and 1 nM of β-estradiol (Sigma-Aldrich), and HEK293T cell line was grown in DMEM (Gibco) supplemented with 10% of heat inactivated fetal bovine serum (Gibco). Cell lines were banked in multiple aliquots on receipt to reduce risk of phenotypic drift and were used for a maximum of 15 passages. All cells have been cultured at 37 °C with 5% CO₂ atmosphere.

### Generation of PDXs

Tumor pieces of 30–60 mm³ obtained from patient primary tumors or metastatic lesions at time of biopsy were immediately implanted into the mammary fat pad (surgery samples) or the lower flank (metastatic samples) of 6-week-old female NOD.Cg-*Prkdc^{scid}Il2rg^{tm1Wjl}/SzJ* mice (*M. musculus*, Charles Rives). Mice were housed in air-filtered flow cabinets with a 12-hours light cycle at 18–23 °C, 40–60% of humidity and food and water *ad libitum*. Mice were continuously supplemented with 1 μmol/L 17β-estradiol (Sigma-Aldrich) in their drinking water, an amount shown to be sufficient to reach serum levels and uterine growth in ovariectomized female mice similar to the ones obtained with other mechanism of 17β-estradiol supplementation. Upon growth of the engrafted tumors, a tumor piece was implanted into the lower flanks of new recipient mice for the model perpetuation. In each passage, flash-frozen and formalin-fixed paraffin embedded (FFPE) samples were taken for genotyping and histological studies. STG201 was generated in CRUK/UCAM as previously reported[21] and PDXs BB3RC31, BB6RC39, BB6RC87, and BB6RC160 were generated in Manchester Breast Center. Both laboratories are members of the EuroPDX consortium.

### Molecular characterization of PDXs

For molecular subtyping, immunohistochemical staining was performed on formalin-fixed paraffin embedded (FFPE) PDXs tissue sections (3 μm). Staining of estrogen receptor (ER), progesterone receptor (PR) and human epidermal growth factor receptor 2 (HER2) were undertaken following the protocol provided by Ventana Medical Systems, Inc. In short, the slides were heated in the oven at 75 °C for 28 min and deparaffinized with EZ prep solution (Ventana Medical Systems). Then, antigen retrieval was performed at slightly basic pH at 95 °C for 56 min. Primary antibodies were incubated for 40 min for ER and HER2 using the Cell Conditioning 1 buffer (CC1; Ventana Medical Systems), and with CC2 buffer for the PR antibody. Finally, the slides were counterstained with Hematoxylin II and Bluing Reagent (Ventana Medical Systems) and mounted with xylol based mounting medium. An investigator blinded to identify the samples quantified the percentage of positively stained cells.

Flash-frozen pieces of tumor xenograft were used for RNA sequencing and PAM50-molecular subtype classification. All the tumor samples used in this study were pieces of patient-derived xenografts. After surgical resection, the tumors were dissected, and a piece was quickly frozen in liquid nitrogen and stored at −80 °C. A frozen tumor specimen was then homogenized in RNAse-free containing lysis buffer and mRNA was prepared by using a PerfectPure RNA Tissue Kit-50 from 5 Prime and protocol. 250 ng of total RNA were used to measure the expression of 50 genes of the PAM50 intrinsic subtype predictor assay and 5 housekeeping genes (*ACTB*, *MRPL19*, *PSMC4*, *RPLPO*, and *SF3A1*) using the nCounter platform (NanoString Technologies). Data was log base2-transformed and normalized to the housekeeping genes using the nSolver 4.0 software and the script developed by[61] R (3.4.3) software. All PDX tumors were assigned to an intrinsic molecular subtype of breast cancer (Luminal A, Luminal B, HER2-enriched, Basal-like or Normal-like) using the PAM50 subtype predictor.

Flash-frozen pieces of tumor xenografts were used for targeted exome sequencing by the MSK-IMPACT™ (Integrated Mutation Profiling of Actionable Cancer Targets), a hybridization capture-based next-generation sequencing assay for targeted deep sequencing designed to capture all protein-coding exons and selected introns of 410 commonly implicated oncogenes, tumor suppressor genes, and members of pathways deemed actionable by targeted therapies[25]. Barcoded sequence libraries were prepared using 100–250 ng genomic DNA (Kapa Biosystems) and combined into equimolar pools of 13–21 samples. The captured pools were subsequently sequenced on an Illumina HiSeq 2000 as paired-end 100-base pair reads, producing a median of 588-fold coverage per tumor. Sequence data were demultiplexed using CASAVA, and reads were aligned to the reference human genome (hg19) using BWA and post-processed using the Genome Analysis Toolkit (GATK) according to GATK best practices. MuTect and GATK were used to call single-nucleotide variants and small indels, respectively. Candidate mutations were manually reviewed using the Integrative Genomics Viewer (IGV) to eliminate likely false positive calls. Because matched normal DNA was not available, tumors were compared to a pool of unmatched normal samples to eliminate common polymorphisms and systematic sequencing artifacts.

## In vivo experiments

To evaluate the sensitivity to the different targeted therapies each PDX was implanted subcutaneously in six-week-old female athymic nude HsdCpb:NMRI-Foxn1nu mice (*M. musculus*, Janvier) or NOD.Cg-*Prkdc*$^{scid}$*Il2rg*$^{tm1Wjl}$*/SzJ* mice (*M. musculus*, Charles Rives) and supplemented with 1 μmol/L 17β-estradiol (Sigma-Aldrich) in their drinking water. Mice were housed at 18–23 °C with 40–60% of humidity and 12 light/12 dark cycle. Food and water were accessible all time. Upon xenograft growth, tumor-bearing mice were randomized into treatments group with tumors ranging 100–300 mm³ (for drug efficacy experiments) or ~500 mm³ (for short-term pharmacodynamic experiments). Ribociclib was administrated by oral gavage once daily, six days/week, at 75 mg/kg (total daily dose) dissolved in distilled water 0.5% hydroxymethyl cellulose. Alpelisib was dosed with the same schedule at 35 mg/kg dissolved in distilled water 0.5% methylcellulose. The combination was administrated with one-hour delay between ribociclib (first) and alpelisib (second). Fulvestrant was administered subcutaneously twice weekly 10 mg/mice dissolved in peanut-oil, letrozole by oral gavage three times per week (1 day on and 1 day off) 20 mg/kg dissolved 0.5% methylcellulose and trastuzumab by intraperitoneal injection twice weekly 10 mg/kg in PBS.

Tumor growth was measured bi-weekly blinded to the treatment effect with a caliper the first day of treatment and to day 35 (for the efficacy assays of ribociclib, alpelisib and their combination), day 15 (for the efficacy assays of fulvestrant and trastuzumab) or day 12 (pharmacodynamic assays). Mice weights was recorded twice weekly.

The tumor volume was calculated using the ellipsoid formula: $V = (\text{length} \times \text{width}^2) \times (\pi/6)$. Mice were euthanized when tumors reached 1500 mm³ or in case of severe weight loss, in accordance with institutional guidelines. All the efficacy experiments contained an untreated control arm with a percentage of change in tumor growth superior to 20% from the initial volume. The antitumor activity was determined by comparing tumor volume at last day of treatment to its baseline (day1): % tumor volume change = $(V_{35\text{days}} - V_{\text{baseline}})/V_{\text{baseline}}$ x100. The antitumor response of subcutaneous implants was classified according to the relative change in tumor volume upon treatment (similar to the Response Evaluation Criteria in Solid Tumors (RECIST) and labeled as mRECIST[20,22]. Complete response (CR) was set as best response ≤ −95%; partial response (PR) as −95%<best response ≤ −30%; stable disease (SD) as −30%<best response ≤ +20%; and progressive disease (PD) as best response > +20%. The models that displayed a preclinical benefit from ribociclib (SD, PR, and CR) were categorized as ribociclib-sensitive. All PD models were categorized as ribociclib-resistant. At the end of the experiment, animals were euthanized using $CO_2$ inhalation. Tumor volumes are plotted as mean values ± SEM. Ribociclib-sensitive models were chronically treated with ribociclib until progression to acquire ribociclib-resistance. Tumor growth was measured once per week and mice weights were recorded twice per week. If mouse welfare was compromised before tumor progression, tumors were harvested and implanted into another recipient mouse. Dosing schedule was reinitiated 10 days post-surgery and lasted until progression.

## PDC ex vivo cultures

Patient-derived tumor cells (PDC) were isolated from PDX through combination of mechanic disruption and enzymatic disaggregation[21]. Briefly, PDX tumors not bigger that 500mm³ were freshly collected in DMEM/F12/HEPES (GIBCO) after surgery resection, minced using sterile scalpels and dissociated for a maximum of 90 min in DMEM/F12/HEPES (GIBCO), 1 mg/ml collagenase (Roche), 100 u/ml, hyaluronidase (Sigma-Aldrich), 5% BSA (Sigma-Aldrich), 5 μg/ml Insulin and 50 μg/ml gentamycin (GIBCO). This was followed by further dissociation using trypsin (GIBCO), dispase 5 mg/ml (StemCell technologies) and DNase 1 mg/ml (Sigma-Aldrich). Red blood cell lysis was done by washing the cell pellet with 1X Red Blood Cell (RBC) Lysis Buffer containing ammonium chloride (Invitrogene). Then, cells were resuspended in MEGM™ Mammary Epithelial Cell Growth Medium Bulletkit™ (LONZA) supplemented with 2% of fetal bovine serum and 10 μM of ROCK inhibitor (Sigma-Aldrich). To test drug antiproliferative responses and for immunoblotting analysis, cells were seeded on collagen-enriched matrix Corning® Matrigel® growth factor reduced (GFR) basement membrane matrix (Corning, INC) at 2 × 10⁵ cells/ml in 8 well-chamber slides (NUNC) or 1 × 10⁶ cells/ml in 6 well plates (BD Biosciences), respectively. For antiproliferative analysis, PDCs were treated with vehicle (DMSO), 1 μM of ribociclib, 500 nM palbociclib, 2.5 μM alpelisib, 100 nM fulvestrant, 20 μM P18IN003 or the combinations and cultured at 37 °C in 5% of $CO_2$. Medium and treatments were refreshed every 2–3 days. For immunoblotting analysis, cells were treated for 24 h. Matrigel® was melted in PBS-EDTA 1 mM on ice for 20 min, the spheroids were collected into a conical tube and centrifuged at 450 × g for 5 min at 4 °C. Pellets were stored at −20 °C until protein lysates were prepared for immunoblotting analysis (see below).

## Analysis of PDCs area

Cell suspensions generated from a 500 mm³ PDX were plated in duplicated at 60.000 cells/well into 8 well-chambers slides. Drugs and vehicle (DMSO) were added after 24 h. To quantify the drug response in PDCs, representative bright field pictures of each well were obtained 7 days post-treatment and normalized against untreated (vehicle). A minimum of three different biological replicates (different tumors)

from each model were assayed. For bright field images analysis ImageJ (http://rsb.info.nih.gov/ij/) was used. Two representative areas of single spheroids were manually quantified individually from at least two independent wells. The mean spheroid area for every treatment was calculated and normalized to untreated controls (vehicles). Relative mean spheroid areas for every treatment condition and the ± SEM were plotted.

### Analysis of S-phase entry cells by EdU incorporation

Cell suspensions generated from a 500mm³ tumors were plated in duplicated at 60,000 cells/well into 8 well-chambers slides. After 24 h, drugs and vehicle (DMSO) as well as 10 μM of 8-ethynyl–2′-deoxyuridine (EdU) were added and the cells were incubated for 2 days. EdU staining was performed using the Click-iT™ EdU Alexa Fluor™ 488 Imaging kit (ThermoFisher Scientific) adapting the manufacturer's instructions. Briefly, the cells were fixed with 3.7% paraformaldehyde for 15 min and permeabilized with 1% Triton X-100 for 20 min, all at room temperature. After 1 h of 5% BSA in PBS blocking, cells were incubated with the Click-iT™ reaction cocktail and primary antibodies (mouse Vimentin 1:500 or human CK18 1:100) overnight at room temperature. The following day, cells were washed 3x with 3% BSA in PBS and incubated with secondary antibodies for 1 h at room temperature. Finally, cells were washed 3x with 3% BSA in PBS, mounted with Prolong™ Antifade Reagent Mountant with DAPI (Molecular Probes) and stored at −20 °C until analysis.

Confocal microscopy analysis was carried out using the Nikon confocal microscope C2⁺equipped with LU-N4S laser unit and the NIS-Elements software (5.10) was used for capturing representative images of spheroids. Number of both DAPI positive and EdU positive cells in each spheroid was manually obtained using ImageJ 1.51 (http://rsb.info.nih.gov/ij/). The percentage of Edu positive cells per spheroid was calculated and the mean of every treatment was relativized to the untreated (vehicle). Relative percentage of S-phase entry cells and ± SEM were plotted.

### Molecular modeling of the complex between p16 and P18IN003

We constructed a structural model of the complex between p16 and a known inhibitor P18IN003 using computational methods of docking and molecular dynamics simulations. p16 has four Ankryin repeats and we used the only available apo structure of p16, an NMR structure (PDB: 2A5E), which is very similar to the crystal structure of p16 bound to CDK6 (PDB: 1BI7, root mean squared deviation of <1 Å, confined largely to the loop regions). An homology model[62] was constructed to model the interactions between p16 and CDK4 based on the p16-CDK6 crystal structure, given that similarity between CDK4 and CDK6 is 81%. This crystal structure was then used to identify the region of interaction between p16 and CDK4 and this region was used to define a binding pocket on the surface of the NMR structure of apo p16 (a similar method was used to identify inhibitors of p18[31]) to which the inhibitor P18IN003 was docked. For docking, the 3D structure of the inhibitor P18IN003 was built using the Maestro module and minimized using the Macromodel module, employing the OPLS-2005 force field, in the program Schrodinger 12.0. The minimized P18IN003 inhibitor was docked into the binding pocket of p16 defined above with the program Glide using standard docking protocols[63]. Out of the top 10 lowest energy poses of the binding of P18IN003 to p16, 8 poses of the inhibitor were very similar to each other and so we chose the top pose and subject the complex to further refinement using molecular dynamics (MD) simulations. The simulations were carried out using the Amber (18) program, using protocols that we have shown to be successful in previous studies[63]. The partial charges and force field parameters for P18IN003 were generated using the Antechamber module in Amber. All atom versions of the Amber ff14SB force field and the general Amber force field (GAFF) were used for the protein and the inhibitors respectively. Simulations were carried out for 100 ns in

triplicates at 300 K using standard protocols[63]. Simulation trajectories were visualized using VMD (1.9.1) and figures were generated using Pymol (2.3.2).

### Immunohistochemistry (IHC) and image analysis

PDX and patient tumors were fixed immediately after excision in 10% buffered formalin solution for a maximum of 24 h at room temperature before being dehydrated and paraffin embedded (FFPE). IHC was performed on FFPE tissue sections (3 μM). For pRb, cyclin D1 and cyclin E1, sections were dewaxed, rehydrated and antigen retrieved using a microwave at maximum power (Whirlpool JT479/WH) in EDTA 1 mM pH 8 (pRb), citrate pH 6 (cyclin D1) or pH 9 (cyclin E1) for 20 min. After peroxide blocking, the slides were stained with the corresponded primary antibodies at dilution 1:100 (pRb and cyclin D1) or 1:300 (cyclin E1), then with ready-to-use Mouse (pRb and cyclin E1) or Rabbit (cyclin D1) EnVision™ + System-HRP labeled polymer and finally with Liquid DAB + Substrate Chromogen System (DAKO). Harris´ hematoxylin was used to counterstain the nuclei and mounted with xylol based mounting medium. Positive and negative controls were run along the tested slides per each marker.

Staining of p16, phospho-pRb (Ser807/811), and Ki67 was undertaken following the protocol provided by Ventana Medical System, Inc. Briefly, the slides were heated in the instrument at 75 °C for 28 min and deparaffinized with EZ prep solution (Ventana Medical System). Then, the antigen retrieval was performed at slightly basic pH at 95 °C using the Cell Conditioning 1 buffer (CC1; Ventana Medical System) followed by staining with anti-rabbit HQ (KI67; 1:500) or anti-mouse HR (p16 and phospho-pRb (Ser807/811) ready-to-use and 1:600, respectively) and anti-HQ-HRP and DAB staining (Roche). The slides were counterstained with Hematoxylin II and bluing reagent (Ventana Medical System) and mounted with xylol based mounting medium.

A pathologist scored the different proteins expression in each sample. Total pRb and p16 were scored semi-quantitatively onto life images with very strong (4+), strong (3+), moderate (2+) weak (1+) or negative staining (0). Allred scores of cyclin D1 and cyclin E1 were calculated from 0 to 8 taking into account the percentage of positive cells (0 to 5) plus the staining intensity (0 to 3). The percentage of cells with nuclear Ki67 or phospho-pRb (Ser807/811) staining was quantified in samples at baseline and after ribociclib treatment from pharmacodynamic experiments.

### In vitro cell line assessment

For half-maximal inhibition concentration (IC$_{50}$) analysis, cells ($2 \times 10^3$/well) were seeded into 96-well plates (BD Bioscience) and, after 24 h, were treated for 6 days with different concentrations of ribociclib, fulvestrant or the combination of drugs. The treatments and media were refreshed every 3 days. Cell proliferation was measured at day 0 and day 6 by fixing with 4% glutaraldehyde (MERCK) in PBS, staining with 0.1% of crystal violet (Sigma-Aldrich) in methanol, solubilizing with 10% of acetic acid in PBS and measuring the absorbance of each well at 560 nm. Values at day 6 were normalized with values at day 0, relativized to controls (vehicle-treated cells) and plotted as the percentage inhibition against the log concentration of ribociclib. IC$_{50}$ was determined using a sigmoidal regression model and was defined as the concentration of drug required for a 50% reduction in growth. Each experiment was repeated three times with three technical replicates.

For biochemical analysis, cells ($1.5 \times 10^6$/well) were seeded into p100 dishes (BD Bioscience) and, the following day, were treated for 24 h or 5 days with 0.5 μM, 1 μM or 1.5 μM ribociclib, 100 nM fulvestrant or the combination of drugs. Next, cells were harvested and prepared for immunoblotting analysis (see below). T47D-p16 cells were incubated with 1 μg/ml doxycycline 48 h prior to add the treatments for inducing p16 expression.

For competition experiments cells ($5.0 \times 10^5$/well) were seeded into p100 dishes (BD Bioscience). 5% of MCF7- or T47D- cells

overexpressing p16 or cyclin D1 ($2.5 \times 10^4$/well) were mixed with 95% of MCF7- or T47D-MOCK cells ($4.75 \times 10^5$/well), seeded together and, the following day, were treated with 1 µM ribociclib for 0, 5 or 14 days. Cells were harvested and prepared for immunoblotting analysis (see below).

## Immunoblotting

Both flash-frozen tumor pieces from pharmacodynamic assays and harvested cells were lysed in ice-cold buffer containing Tris-HCl pH7.8 20 mmol/L, NaCl 137 mmol/L, EDTA pH 8.0 2 mmol/L, NP40 1%, glycerol 10%, supplemented with NaF 10 mmol/L, Leupeptin 10 mg/mL, $Na_2VO_4$ 200 mmol/L, PMSF 5 mmol/L, and Aprotinin (Sigma-Aldrich). Tissue homogenization was performed on ice with a POLYTRON® system PT 1200 E (Kinematica). Lysates were centrifuged at $18,000 \times g$ 4 °C for 10 min and the supernatants were collected. Protein concentration was calculated using DCTM Protein Assay (Bio-Rad). A total of 30 µg of protein were separated on 12% SDS-PAGE acrylamide gels at 100 V and transferred to nitrocellulose membrane for 1.5 h at 100 V. Membranes were blocked for 1 h in 5% milk in Tris-buffered saline (TBS)-Tween and then hybridized using the corresponding primary antibodies in 5% BSA TBS-Tween. All primary antibodies were used at dilution 1:1000. Mouse and rabbit horseradish peroxidase (HRP)-conjugated secondary antibodies (1:2000; GE Healthcare) were diluted in 5% milk in TBS-Tween and proteins were detected with Immobilon Western Chemiluminescent HRP substrate (Millipore). Images were captured with FUJIFILM LASS-4000 camera system and quantified with ImageJ (1.51). Uncropped and unprocessed scans of the most important blots are supplied in the Source Data file.

## Lentiviral infection

Lentiviral infection was done following the manufacturer´s indications (Sigma-Aldrich). Briefly, 293FT cells were used for the production of the virus. 293FT cells ($5 \times 10^6$) were transfected with lentivirus and packaging (gag-pol, vsvg, rev) plasmids (Addgene) using polyethyleneimine (Sigma-Aldrich) in a DNA-PEI ratio of 1:3. Viral production was induced by adding 10 mM Na Butyrate the following day. Virus were harvested 48 h post transfection. p16-INK4A and cyclin D1 lentiviral stocks were tittered using colony forming assay (Hela cells). MCF7 and T47D cells were infected with doxycycline-inducible pLX401-INK4A (MOI 1:2) for overexpressing p16 or pLenti-CCND1 (MOI 1:2) and the control plasmid (pLenti-tGFP) for overexpressing cyclin D1. 8 µg/ml of polybrene (Sigma-Aldrich) were added, plates were centrifuged 1 h at $450 \times g$ at 37 °C to improve the infection and incubated overnight. For p16-INK4 cell selection, 2 µg/ml of puromycin (Invitrogene) were added to the cultured media 48 h post-infection. After 5 days, all cells in the control plate (non-infected cells) were dead and the concentration of puromycin was reduced to 1 µg/ml (maintenance dose). For p16 expression, 1 µg/ml of doxycycline (Sigma-Aldrich) was added to the culture media 48 h before treatments were added.

## Real Time-qPCR ready custom panels

RNA was extracted from flash-frozen control and ribociclib-treated PDX samples (15–30 mg) by using the PerfectPure RNA Tissue kit (five Prime). The purity and integrity were assessed by the Agilent 2100 Bioanalyzer system, and cDNA was obtained using the PrimeScript RT Reagent kit (Takara). Quantitative RT-PCR was performed in the LightCycler® 480 (Roche) using LightCycler® 480 Probes Master (Roche) and ready-to-use custom 384-plates panels containing predesigned human specific primers and UPL Probes for each gene (Supplementary Table 5). Tbd probes were designed and tested specifically for this assay. The comparative CT method was used for data analysis, in which geoNorm algorithms were applied to select the most stably expressed housekeeping genes (GAPDH and ACTB) and geometric mean values were calculated to obtain normalized CT values[64]. qPCR assay IDs are shown in Supplementary Table 5.

## Statistical analysis

GraphPad Prism (6.00) for Windows was used for statistical analysis. A bootstrap resampling procedure ($n = 2000$) was used to calculate the standard error in the percentage of change in tumor volume relative to untreated. D'Agostino-Pearson omnibus test was performed to check the normality assumption in all comparative studies. If the null hypothesis of normality was not rejected, we assumed Gaussian distribution of the samples, but if the sample size was too small or the hypothesis was rejected, we did not assume it.

For the comparative experiments of biomarkers between ribociclib and vehicle groups, we used paired t-test or Wilcoxon signed-rank test, as appropriate after checking normality assumption. For the comparative between ribociclib-sensitive (S) and -resistant (R) samples, we used an unpaired t-test or Mann-Whitney U test, as appropriate after checking normality assumption. Adjustment for multiple testing was performed in each biomarker by controlling the false discovery rate at 5% according to the Benjamini and Hochberg method.

Univariate logistic models were used to obtain odds ratios (OR) of studied biomarkers in PDXs. To quantify the level of association between a qualitative factor and response we calculated accuracy, sensitivity, specificity, positive and negative predictive values (PPV and NPV). The optimum cut-off points established (p16 high ≥ 2, pRb low ≤ 2 and Cyclin E1/D1 high > 4/6) in this study were selected by the Youden index, which maximizes the sum of the sensitivity and specificity in each biomarker analyzing the ROC curve.

For the ABC-POP trial we performed a logistic regression of biomarkers and KI67 percentage, univariate logistic regressions to estimate the odds ratio for a standard deviation change in continuous H-score biomarkers levels of p16, pRb and cyclin D1 and absolute KI67 response and a multivariable logistic regression analysis. For the comparative between ribociclib-sensitive (S) and -resistant (R) metastatic patient samples, we used an unpaired t-test or Mann-Whitney U test, as appropriate after checking normality assumption. We also applied univariate logistic regressions to estimate the odds ratio. The analysis was performed with R (4.0.3).

For analyzing the correlation between double genetic hit in RB1 locus and treatment with CDK4/6i, genomic data of metastases from 800 BC was obtained from the Hartwig Medical Foundation (HMF[36]); (DR-110). Patients with 'None', 'Other' or 'Unknown' treatment were filtered out, yielding 582 patients who received known treatments. All metastatic samples bearing single base substitutions and indels causing frameshift variants, stop gained variants, splice acceptor variants, splice donor variants, start lost variants, stop lost variants, missense variants, inframe deletions or inframe insertions affecting the RB1 gene were identified. Metastatic samples with minor and major allele copy number of the genomic region containing the RB1 locus smaller than 0.01 were deemed to carry homozygous deletion of the RB1 gene, while those with minor allele copy number of this genomic region smaller than 0.01, but greater major allele copy number were deemed to carry a heterozygous deletion of RB1. We used multivariable logistic regression to assess the association between the alteration status of the metastatic breast tumor and the likelihood probability that the patient received CDK4/6i. Two logistic regression formulas modeling different interactions between the variables (mutation and deletion) were applied.

## Reporting summary

Further information on research design is available in the Nature Research Reporting Summary linked to this article.

## Data availability

The annotated DNA sequencing data of the PDX models generated in this study is provided in the Source Data file. Access to the raw sequencing data can be made available for academic research only

from Dr Violeta Serra (vserra@vhio.et) and after completing a Data Transfer Agreement with Memorial Sloan-Kettering Cancer Center (New York, USA). Genomic and clinical data of metastatic breast cancer patients from the Hartwig Medical Foundation (HMF) cohort and ABC-POP trial analyzed used in this manuscript were previously published and requested to the corresponding main authors [doi: 10.1038/s41586-019-1689; 10.1016/j.annonc.2020.08.283]. Clinical data generated in this study from metastatic breast cancer patients treated at Vall d´Hebron Hospital are provided in the Source Data file. Clinical data shown in Fig. 6C and supplementary 2C of this manuscript were obtained from cBioportal [https://www.cbioportal.org/]. The 3D protein and compound structures employed in this manuscript are deposited in the PDB (Protein Data Bank) archive: 2A5E, 1BIN and P18IN003 [https://pubchem.ncbi.nlm.nih.gov/compound/9994705]. Source data are provided with this paper.

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

## Acknowledgements

The authors are grateful to the patients who consented for the use of their tumors and the personnel involved in sample collection from the Breast Surgical Unit, Breast Cancer Center, the Department of Radiology Vall d'Hebron University Hospital; the Molecular Oncology Group at Vall d'Hebron Institute of Oncology (VHIO). Emmanuelle di Tomaso from Novartis Pharmaceuticals Corporation kindly provided LEE011 and BYL719 for in vivo and ex vivo work. Genentech, Inc. kindly provided GDC-0032 for in vivo and ex vivo work. We are also grateful to Hong Shaoh at the MSKCC and the Molecular Biology Group at Vall d´Hebron University Hospital for providing technical support with PDXs sequencing and immunostainings, respectively. This manuscript was edited by Life Science Editors. Artwork by vector.me. The ABC-POP received funding from the Breast Cancer Research Foundation and research grants from Pfizer and Eli-Lilly, respectively (to M.A.). This publication and the underlying research are partly facilitated by Hartwig Medical Foundation and the Center for Personalized Cancer Treatment (CPCT) which have generated, analyzed, and made available data for this research. We acknowledge Novartis, Genentech, the GHD-Pink program, the FERO Foundation and the Orozco Family for supporting this study [to V.S.]. This study has been supported by the Susan G. Komen

Foundation (CCR15330331) and by the Catalan Agency AGAUR (2017 SGR 540) [to V.S.]. V.S. received funds from the Instituto de Salud Carlos III: grants PI13/01714, CP14/00228, MV15/00041, CPII19/00033 and PI20/00892. M.P. received a Juan de la Cierva Grant from the Ministerio de Economía y Competitividad (FJCI-2015-25412), L.Mo. a grant from FI-AGAUR (2019 FI_B 01199), F.B-M. a grant from the Fundación Científica Asociación Española Contra el Cáncer (AECC_Postdoctoral17-1062) and M.S-G, a Marie Slodowska-Curie Innovative Training Networks PhD fellowship (H2020-MSCA-ITN-2015_675392). This work was supported by Breast Cancer Research Foundation (BCRF-19-08), Instituto de Salud Carlos III Project Reference number AC15/00062 and the EC under the framework of the ERA-NET TRANSCAN-2 initiative co-financed by FEDER, Instituto de Salud Carlos III (CB16/12/00449 and PI19/01181) and Asociación Española Contra el Cáncer (to J.A.). R.B.C. laboratory is supported by Breast Cancer Now (grant numbers: MAN-Q1 and MAN-Q2), NIHR Manchester Biomedical Research Centre (IS-BRC-1215-20007) and EdiREX Horizon 2020 grant No.731105. The xenograft program in the C.C. laboratory is supported by Cancer Research UK and also received funding from an EU H2020 Network of Excellence (Euro-CAN). This work has been supported by NIH grants P30 CA008748 and RO1CA190642-01, the CDMRP grant BC171535P1, and the Breast Cancer Research Foundation [to M.S.]. A.P. received funds from Instituto de Salud Carlos III—PI16/00904 and PI19/01846, Breast Cancer Now—2018NOVPCC1294, Breast Cancer Research Foundation-AACR Career Development Awards for Translational Breast Cancer Research 19-20-26-PRAT, Fundació La Marató TV3 201935-30, the European Union's Horizon 2020 research and innovation program H2020-SC1-BHC-2018-2020. IRB Barcelona is a recipient of a Severo Ochoa Centre of Excellence Award from the Spanish Ministry of Economy and Competitiveness (MINECO; Government of Spain) and is supported by CERCA (Generalitat de Catalunya). C. S. Verma reports grants from MSD International and grants from Ipsen outside the submitted work.

## Author contributions

M.P.: collection and/or assembly of data, data analysis and interpretation, and manuscript writing. L.Mo., S.K., M.T. H.-A., A.O., M.S.-G., O.R., M.G., and J.G.: collection and/or assembly of data, data analysis and interpretation. G.V., A.G.-P., F. B.-M., N.I., S.M., N.L.-V., and R.D.: statistical design and clinical trial data analysis. V.S.: conception and design of the study, financial support, collection and/or assembly of data, data analysis and interpretation, and manuscript writing. J.H and P.N.: Pathology assessment of the biological samples. L.Mi., M.C., A.A., C.Vi., and A.G.-S.: collection and/or assembly of clinical trial and clinical samples data. A.B., F.S., K.L., R.B.C., C.C., J.A., A.P., C.Ve., M.B., M.O., N.C.T., M.S., M.A., and C.S.: data analysis and interpretation. All: interpretation of the data analysis and final approval of the manuscript.

## Competing interests

V.S. received non-commercial research support from Novartis and Genentech. M.B. reported receiving honoraria for speaker activities and advisory role from Pfizer, Novartis and Elli-Lilly and support for travel expenses from Roche and Pfizer. M.O. declares grant/research support (to the Institution) from AstraZeneca, Philips Healthcare, Genentech, Roche, Novartis, Immunomedics, Seattle Genetics, GSK, Boehringer-Ingelheim, PUMA Biotechnology, and Zenith Epigenetics; consultant role for Roche, GSK, PUMA Biotechnology, AstraZeneca, and Seattle Genetics; and has received honoraria from Roche, Seattle Genetics, and Novartis. G.V. reported receiving honoraria for speaker activities from MDS and advisory role from Astrazeneca. F.S. is employee of Novartis. K.L. is employee of Genentech. C.C. is a member of AstraZeneca's External Science Panel, of Illumina's Scientific Advisory Board, and is a recipient of research grants (administered by the University of Cambridge) from AstraZeneca, Genentech, Roche and Servier. J.A. has received research funds from Roche, Synthon, Menarini, and Molecular Partners and consultancy honoraria from Menarini. A.P. reports that his institution received research funding from Nanostring Technologies, Roche and Novartis and reports consulting and lecture fees from Nanostring Technologies, Roche, Novartis, Pfizer, Oncolytics Biotech, Amgen, Elli-Lilly, MSD and PUMA. P.N. has consulted for Bayer, Novartis, and MSD and received compensation. R.D. is on advisory role of Astra-Zeneca, Roche and Boehringer-Ingelheim and has received speaker's fees from Roche, Symphogen, IPSEN, Amgen, Servier, Sanofi, MSD, and research support from Merck. M.S. is on the scientific advisory board of Menarini Ricerche and the Bioscience Institute, has received research funds from Puma Biotechnology, Daiichi-Sankio, AstraZeneca, Targimmune, Immunomedics and Menarini Ricerche, and is a cofounder of Medendi.org. M.A. received a research grant from Eli-Lilly, honoraria from Novartis, Astrazeneca, Seattle Genetics, Abbvie and Pfizer and travel grants from Novartis, Roche, Pfizer. S.M. has provide punctual statistical advice to IDDI and Janssen Cilag and participated to data and safety monitoring committees of clinical trials (Hexal, Steba, IQVIA, Roche, Sensorion, Biophytis, Servier, Yuhan), outside the submitted work. C.S. has served as consultant, participated in advisory boards or received travel grants from AstraZeneca, Celgene, Daiichi Sankyo, Roche, Genomic Health, Merck, Sharp and Dhome España S.A., Novartis Odonate Therapeutics, Pfizer, Philips He. C.Ve. & S.K. are founders of Sinopsee Therapeutics and Aplomex; neither company has any conflict with the current work. VHIO has had funding (paid directly to the Institution) from AstraZeneca, Daiichi Sankyo, Eli Lilly and Company, Genentech, Immunomedics, Macrogenics, Merck, Sharp and Dhome España S.A., Novartis, Pfizer, Piqur Therapeutics, Puma, Roche, Synthon and Zenith Pharma. The remaining authors declare no competing interests.

## Additional information

[1]Experimental Therapeutics Group, Vall d'Hebron Institute of Oncology, Barcelona, Spain. [2]Breast Cancer and Melanoma Group, Vall d'Hebron Institute of Oncology, Barcelona, Spain. [3]Department of Medical Oncology, Hospital Vall d'Hebron, Barcelona, Spain. [4]Oncology Data Science Group, Vall d'Hebron Institute of Oncology, Barcelona, Spain. [5]Institute for Research in Biomedicine (IRB Barcelona), Barcelona, Spain. [6]Research Program on Biomedical Informatics, Universitat Pompeu Fabra, Barcelona, Spain. [7]Translational Genomics and Targeted Therapies in Solid Tumors, August Pi i Sunyer Biomedical Research Institute (IDIBAPS), Barcelona, Spain. [8]Service de Biostatistique et d'Epidémiologie, Gustave Roussy, Villejuif, France. [9]Oncostat U1018, Inserm, University Paris-Saclay, Villejuif, France. [10]Bioinformatics Institute (A*STAR), Singapore, Singapore. [11]Medica Scientia Innovation Research (MedSIR), Barcelona, Spain. [12]The Breast Cancer Now Research Centre, London, UK. [13]Preclinical Modelling of Pediatric Cancer Evolution Group, The Institute of Cancer Research, London, UK. [14]Translational Molecular Pathology, Vall d'Hebron Institute of Research (VHIR), Barcelona, Spain. [15]Novartis Pharmaceuticals, East Hanover, NJ, USA. [16]Genentech, Inc., South San Francisco, California, USA. [17]Breast Biology Group, Manchester Breast Centre, Manchester, UK. [18]Cancer Research UK, Cambridge, UK. [19]CIBERONC, Vall d'Hebron Institute of Oncology, Barcelona, Spain. [20]Growth Factors Laboratory, Vall d'Hebron Institute of Oncology, Barcelona, Spain. [21]Department of Biochemistry and Molecular Biology, Universitat Autònoma de Barcelona, Barcelona, Spain. [22]IMIM (Hospital del Mar Medical Research Institute), Barcelona, Spain. [23]Institució Catalana de Recerca i Estudis Avançats (ICREA), Barcelona, Spain. [24]University of Barcelona, Barcelona, Spain. [25]Department of Medical Oncology, Hospital Clinic, Barcelona, Spain. [26]SOLTI Breast Cancer Research Group, Barcelona, Spain. [27]Department of Oncology, IOB Institute of Oncology, Barcelona, Spain. [28]Molecular Oncology Group, Vall d'Hebron Institute of Oncology, Barcelona, Spain. [29]School of Biological Sciences, Nanyang Technological University, Singapore, Singapore. [30]Department of Biological Sciences, National University of Singapore, Singapore, Singapore. [31]Departments of Pathology and Human Oncology and Pathogenesis Program, Memorial Sloan-Kettering Cancer Center, New York, USA. [32]Department of Medical Oncology, Gustave Roussy, Villejuif, France. [33]Inserm Unit U981, Villejuif, France. ✉e-mail: vserra@vhio.net

