## [Peer Review File · Nature Communications]

High p16 expression and heterozygous *RB1* loss are biomarkers for CDK4/6 inhibitor resistance in ER⁺ breast cancerReviewers' comments:

Reviewer #1 (Remarks to the Author):

Summary: The study by Palafox and colleagues is focused on the markers of response to CDK4/6 inhibitors in breast cancer PDX models and clinical specimens. Initially, a panel of PDX models are used to evaluate the response to ribociclib which illustrates a diversity of responses, which are presented as superior to either endocrine therapy or trastuzumab. Molecular and immunohistochemistry analyses performed indicate that high-levels of p16 are associated with resistance, the relationship of RB, Cyclin E1 and Cyclin D1 are not directly associated with response as single markers, but can further refine the predictive power. These data are recapitulated in organoid cultures. Subsequent overexpression studies in cell lines show that either p16 overexpression or cyclin D1 expression promote a degree of resistance to ribociclib. In clinical specimens high p16 associates with resistance, while in the context of adaptive resistance there is conversion from heterozygous loss to deletion of RB.

Critique: This study provides a wealth of information relative to the response/resistance to CDK4/6 inhibition. While overall rigorously performed there are some concerns that diminish enthusiasm that are enumerated below.

1. The criteria used for "responsiveness" is mysterious, relative to the variable growth rates of the PDX models. Notably there is a bias in the cutpoint for slower growing tumors.
2. There is generally a reciprocal relationship between p16 and RB. This relationship is not clearly articulated in the data that is presented. Furthermore, RB deficient tumors often express little cyclin D1. The presentation in Figure 2 makes it challenging to appreciate more than a single marker. RB loss and high-levels of p16 are already known to indicate resistance to CDK4/6 inhibition in preclinical models.
3. Relative to the spheroid cultures, it is unclear how this adds beyond the PDX studies and the difference in growth rate is relatively modest based on the graph of relative spheroid area.
4. The interpretation relative to the expression of p16 is difficult to follow and not necessarily accurate. That p16 may compete for binding is possible, but p16 is a potent inhibitor of CDK4/6. Thus, by growing the cells in the presence of p16 ostensibly the cells have lost the requirement for CDK4/6? This can be tested by determining if the cells that are expressing p16 are sensitive/insensitive to the depletion of CDK4/6 or other methods (Fassl et al., 2020).
5. Multiple other groups have evaluated p16/CDKN2A in different clinical specimens and failed to observe predictive power. This could be due to the nature of the assays or the specimens utilized. However, this limitation should be discussed in more detail and any features that are unique to this study should be contrasted with the other work that generally is using a larger collection of specimens.

Reviewer #2 (Remarks to the Author):

In this study, the authors aim to identify biomarkers of resistance to CDK4/6 inhibition (ribociclib) in a panel of 37 patient-derived tumor models, using genetic, transcriptomic and proteomic approaches. The study claims that 1) high p16 protein levels and heterozygous RB1 loss as novel biomarkers for resistance to CDK4/6i treatment and suggest that 2) CDK4/6i plus PI3Ki may be effective in non-basal-like tumors that progress to CDK4/6i and ET, independently of the PIK3CA or RB1 status. There are several novel findings in this work including that the highest expression levels of p16 may denote CDK4/6 resistance. The authors also propose the integrated biomarker of p16 high, pRb low, cyclin D1 high, and cyclin E1 high expression which showed higher sensitivity (87%) and accuracy (85%) for the detection of ribociclib-resistant models compared to single or binary biomarkers. The combination of PIK3CA and CDK4/6 inhibition is not novel and has been described and is already in clinical trials. However, this work does offer rationale for integration

into the post-CDK4/6 space, elimination of endocrine backbone and de-escalation strategies which could be translationally relevant.

1) PAM50 analysis of 23 PDX models identified 3 basal-like subtypes which were all resistant to ribociclib which is expected. However, this does highlight continue translational relevance of PAM50 which is not currently used in standard practice for early or late stage breast cancer.

2) In this analysis, higher protein levels of p16 levels ($p=0.01$) were detected in resistant PDXs compared to the sensitive ones (Figure 2B). These findings were also validated in a clinical cohort of patients treated with neoadjuvant abemaciclib in the ABC-POP trial ($p=0.008$, Figure 5C). Preclinical overexpression of p16-mediated resistance to CDK4/6 inhibitors in the absence of Rb has been described previously, though this analysis is much more in depth and definitive [1]. The authors should mention that P16 protein expression has also been evaluated large scale biomarker analyses of phase III clinical trials of CDK4/6 inhibitors such as PALOMA 2. In this analysis, P16 IHC was evaluated by Ventana E6H4 anti-p16 antibody using H-score methodology of ≥ 1 as positive. A total of 568 of 666 (85%) baseline tumor tissues from the 666 patients randomized were collected. CDK6, cyclin D, cyclin E, RB, and p16 expression levels do not predict the benefit of palbociclib in combination with letrozole by IHC, FISH, or gene expression analysis [2]. This analysis reports methods of semiquantitative analysis for pRb using H-score methods though cut offs of what is considered high are not clearly presented in the manuscript. High p16 means expression score $\geq 3+$. The vast differences in cut point may partially explain why p16 in clinical trial analysis has not been linked to sensitivity or resistance to CDK4/6 inhibition and could be highlighted.

3) An interesting analysis performed was assessing the value of the integrated biomarker p16 high, pRb low, cyclin D1 high, and cyclin E1 high expression. This composite marker showed higher sensitivity (87%) and accuracy (85%) for the detection of ribociclib-resistant models compared to single or binary biomarkers (Figure 2C). Validation of this integrated biomarker compared to high p16 alone in clinical samples is a logical next step.

4) 432: "The PI3K inhibitor alpelisib sensitizes non-basal like BC PDX to ribociclib". This data is unfortunately not novel and has been shown before. O'Brien et al have shown that direct loss of Rb or loss of dependence on Rb signaling confers cross-resistance to inhibitors of CDK4/6, while PI3K/mTOR signaling remains activated. Treatment with alpelisib completely blocked the progression of acquired CDK4/6 inhibitor-resistant xenografts in the absence of continued CDK4/6 inhibitor treatment in models of both PIK3CA mutant and wild-type ER+/HER2- breast cancer. They also showed that triple combination therapy against PI3K:CDK4/6:ER prevented and/or delayed the onset of resistance in treatment-naive ER+/HER2- breast cancer models[3]. This body of work does give rationale to omission of endocrine therapy in this combination and most importantly de-escalation strategies.

5) 435: Currently, this approach is further being tested in the context of tumors that progress after being treated with CDK4/6i-containing regimens (e.g. BYlieve, NCT03056755). This trial has preliminary results showing that the proportion of patients without disease progression at 6 months (CBR) was 50.4% showing clear efficacy of PIK3CA inhibitors post-CDK4/6 [4].

6) 451: "Because of concerns regarding the safety of the combination of alpelisib plus ribociclib, we conducted de-escalation experiments showing that dose reduction of either ribociclib or alpelisib resulted in similar antitumor activity as the full dose tested in both PDX039 and PDX191 models. Dose reduction of both drugs, however, resulted in an attenuated efficacy". Given that triplet preclinical data has been shown before and is already in clinical trials, I would consider elaborating on this data in the manuscript and highlighting in abstract and or discussion as toxicity of these combinations is a very significant concern. This would strengthen novel translational relevance.

7) 501-507: "Although CDK4/6i are usually administrated in combination with ET, abemaciclib can be used as a single agent for the treatment of patients with ER+ /HER2-negative metastatic BC that progressed on ET and chemotherapy. Also, the data from the TREnd trial suggested that CDK4/6i combined with ET and CDK4/6i monotherapy had similar clinical benefit rates and overall

response rates in post-menopausal women with advanced ET-pretreated ER+/HER2-negative BC. In this sense, our data demonstrates that targeting the CDK4/6 axis is a valid strategy when other targeted therapies against ER or HER2 have failed.” Unclear the direction of this statement. Single agent CDK4/6 inhibition has very little clinical relevance at this point. The TREND trial was quite small and the endocrine therapy backbone continued was that of which patients’ were already resistant to. The success of targeting the CDK4/6 axis after endocrine or trastuzumab resistance has been repeatedly shown in preclinical and clinical trials and not this body of work. I suspect you mean to offer rationale for PI3CA/CDK4/6 inhibition without ET.

Reviewer #3 (Remarks to the Author):

This manuscript employs the power of multiple PDXs to preclinically ask several questions about CDK4/6i. The authors first ask what biomarkers predict resistance to ribociclib, and identify four. RB and Cyclin E have already been reported to be correlated with CDK4/6i. This manuscript confirms these data and adds p16 and cyclin D1 expression. Of these, p16 is validated in cohort studies. The authors then move on to look at RB heterozygosity, I am unsure of what data prompted this. In a tumor with RB heterozygosity, variants with additional mutations showed altered drug sensitivity. This finding is validated in a cohort although its incidence is unclear. Finally, the authors ask if a PI3K inhibitor will add to a CDK4/6i, with or without ER inhibition, and show interesting in vivo data. Strengths of this paper include a panel of 21-23 PDXs, and experiments that investigate clinically important questions. For almost all of the experiments weaknesses exist as listed below. These include a very small number of resistant models, inclusion of triple-neg PDXs which were never expected to matter, inadequate functional experiments, lack of attention to ER mutations, and inadequate discussion of pharmacokinetic variables. If thoroughly fixed, the manuscript may inform future clinical trials.

1. Of the initial 21 PDXs, very few were sensitive to ribociclib (1 CR, 2 PR, 2 SD). This does not supply a robust “N” to distinguish molecular characteristics of responders vs nonresponders. Do the trends hold up if SD is eliminated?
2. Why were the basal PDXs included?
3. Data in Fig 2D should be confirmed with an independent cohort such as METABRIC.
4. The p16 functional data should be performed with two cell lines. Both functional investigations must include in vitro and in vivo data.
5. In Fig. 5C, can the data be broken out by molecular subtype?
6. The metastatic data in Fig 5D, while significant, is based on 2 resistant tumors. This major limitation should be added to Results.
7. It is unclear how the transition from p16 to RB heterozygosity occurred. Fig 6C must list the number of specimens with RB heterozygous loss.
8. The dose de-escalation study in Fig 7 should provide detail on the serum levels of the drugs, versus what is clinically achievable and safe.
9. The issue of ER mutation is not analyzed and is needed.

Reviewer #1 (Remarks to the Author):

Summary: The study by Palafox and colleagues is focused on the markers of response to CDK4/6 inhibitors in breast cancer PDX models and clinical specimens. Initially, a panel of PDX models are used to evaluate the response to ribociclib which illustrates a diversity of responses, which are presented as superior to either endocrine therapy or trastuzumab. Molecular and immunohistochemistry analyses performed indicate that high-levels of p16 are associated with resistance, the relationship of RB, Cyclin E1 and Cyclin D1 are not directly associated with response as single markers, but can further refine the predictive power. These data are recapitulated in organoid cultures. Subsequent overexpression studies in cell lines show that either p16 overexpression or cyclin D1 expression promote a degree of resistance to ribociclib. In clinical specimens high p16 associates with resistance, while in the context of adaptive resistance there is conversion from heterozygous loss to deletion of RB.

Critique: This study provides a wealth of information relative to the response/resistance to CDK4/6 inhibition. While overall rigorously performed there are some concerns that diminish enthusiasm that are enumerated below.

1. The criteria used for “responsiveness” is mysterious, relative to the variable growth rates of the PDX models. Notably there is a bias in the cut-point for slower growing tumors.

We used a criterion for responsiveness that is widely used and closely mimics the response evaluation in the clinic, namely RECIST criteria. This criterion is based on tumor growth inhibition compared to baseline rather than a comparison to the untreated control. This is important because using a comparison to an untreated control can result in statistically significant tumor growth reduction that still leads to disease progression and therefore is not clinically meaningful.

Regarding the bias in the cut-point for slower growing tumors, we ruled out that CDK4/6i-sensitive tumors have statistically slower growth than CDK4/6i-resistant tumors (Figure for Reviewers 1A). This observation rules out a potential bias for slower growing tumors being more sensitive.

For a better understanding, we have modified the results section as follows: *“We then examined the antitumor activity of ribociclib in these 21 PDXs and responses to therapy were classified according to the relative change in tumor volume upon treatment (similar to the Response Evaluation Criteria In Solid Tumors (RECIST) criteria). (...). The degree of response to ribociclib was independent of the tumor growth rate of the untreated tumors, ruling out a potential bias for slower growing tumors being more sensitive.”*

2. There is generally a reciprocal relationship between p16 and RB. This relationship is not clearly articulated in the data that is presented. Furthermore, RB deficient tumors often express little cyclin D1. The presentation in Figure 2 makes it challenging to appreciate more than a single marker. RB loss and high-levels of p16 are already known to indicate resistance to CDK4/6 inhibition in preclinical models.

We have clarified this point in the results section in the statement: *“Two out of 3 ER⁺, basal-like PDX had high p16 levels with concomitant low pRb expression (PDX313, PDX098, Figure 2B), which was expected given the generally reciprocal relationship between p16 and pRb (Figure S2C)[1-3].”* Please note that we have changed the references linked to this statement.

We have verified that, as suggested by the Reviewer, ER⁺ pRb-deficient tumors express low cyclin D1, which is statistically significant at the protein level in the TCGA cohort ($p=0.02$, Figure for Reviewers 1B). Also, we illustrate the biomarker relationships in our models in Figure for Reviewers 1C, where the relationship between pRb deficiency and low cyclin D1 is not observed. To highlight the biomarker relationships in our models, we have changed the position of Figure S2B to Figure 2C.

Regarding the third part of the reviewers' question, *RB1* loss has been widely validated as a CDK4/6i-resistance gene [4-7]. In relation to high-levels of p16, we previously demonstrated that MCF7 cells overexpressing p16 showed an impaired engagement of CDK4 by palbociclib [8]. More recently, the group of Dr. Chandralapaty has been shown that INK4 proteins such as p18 or p16 induce resistance in CDK6-overexpressing tumors by occluding CDK4/6i-binding while

only weakly suppressing ATP binding [9]. To the best of our knowledge, there is a lack of data showing if p16 overexpression is associated with CDK4/6i-resistance in pRb-functional, ER+ breast cancer cell lines. Our study shows that p16-overexpression in cell line models that retain functional pRb leads to CDK4/6i +/- fulvestrant resistance (Figure 4A-C and Figure S3A-C). This is also observed in the multivariable analyses of the ABC-POP trial, where high p16 expression alone was significantly associated with a decreased probability of Ki67-response ($p=0.004$; Figure 5C and Figure S4B).

3. Relative to the spheroid cultures, it is unclear how this adds beyond the PDX studies and the difference in growth rate is relatively modest based on the graph of relative spheroid area.

We agree with the Reviewer that the difference in spheroid growth rate is relatively modest compared to the separation between sensitive and resistant models shown in the *in vivo* experiments. This might be due to shorter time-window utilized for the spheroid cultures. Nonetheless, spheroid cultures have enabled biomarker validation that were identified with the PDX studies, avoiding the use of an incremental number of mice, and we stand to include this data in the manuscript.

4. The interpretation relative to the expression of p16 is difficult to follow and not necessarily accurate. That p16 may compete for binding is possible, but p16 is a potent inhibitor of CDK4/6. Thus, by growing the cells in the presence of p16 ostensibly the cells have lost the requirement for CDK4/6? This can be tested by determining if the cells that are expressing p16 are sensitive/insensitive to the depletion of CDK4/6 or other methods (Fassl et al., 2020).

Cells overexpressing p16 are still partly dependent on CDK4/6, as shown in the ribociclib dose-response and biochemical data shown in Figure 4A, B.

Mechanistically, we previously demonstrated that binding of p16 to CDK4 attenuates palbociclib engagement hampering its kinase activity inhibition [8]. More recently, Dr. Chandarlapaty from MSKCC, NY, has obtained complementary results, shown in his manuscript entitled "*INK4 tumor suppressor proteins mediate resistance to CDK4/6 kinase inhibitors*" [9]. In this manuscript, it has been shown that INK4 proteins such as p18 or p16 induce resistance to CDK4/6i in CDK6-overexpressing tumors. Specifically, INK4 proteins preferentially bind CDK6 over CDK4 and distort the ATP/drug binding pocket of CDK6 in a manner that binding of ATP is favored over palbociclib, inducing drug resistance. These findings are relevant for the clinic because multiple genetic alterations lead to CDK6-mediated resistance (e.g., *FAT1*, *PTEN*). A PROTAC drug to specifically degrade CDK6 was able to overcome CDK4/6i resistance. Complementary, our results further show that targeting the p16-CDK4/6 interaction sensitizes p16-overexpressing tumor cells to CDK4/6i. This evidence has been introduced in the Discussion to strengthen the interpretation relative to p16.

5. Multiple other groups have evaluated p16/CDKN2A in different clinical specimens and failed to observe predictive power. This could be due to the nature of the assays or the specimens utilized. However, this limitation should be discussed in more detail and any features that are unique to this study should be contrasted with the other work that generally is using a larger collection of specimens.

We acknowledge that other groups have evaluated p16/CDKN2A in association with CDK4/6i plus hormone therapy resistance in clinical trial specimens and that the nature of the assays, namely cut-off points could have impacted the results. To the best of our knowledge, p16 protein expression was assessed in the PALOMA-2 trial [10, 11] and in MONALEESA-2 [12, 13] and we used the same assay as in PALOMA-2, namely the Ventana CINtec p16 Histology test (E6H4 antibody).

In PALOMA-2, efficacy of palbociclib plus letrozole was not affected in individuals according to their tumor (i) p16 loss as H-Score<1 nor (ii) p16 high as H-Score>80 (quartile analysis)[11]. Of note, an H-Score of 80 (over 300) would not be representative of p16 overexpression.

In MONALEESA-2, treatment benefit was also maintained irrespective of p16 protein expression. These results were presented during the AACR Annual Meeting in 2017 and showed that no

difference in treatment benefit was achieved according to the mean of p16 expression levels by IHC (please note that abstract and talk presented different cut-offs). As in PALOMA-2, mean values of p16 would not be representative of p16 overexpression.

In addition, we would like to point out that the incidence of the individual alterations associated with CDK4/6i resistance, e.g. *RB1* loss, *CCNE1* overexpression and especially p16 overexpression, is relatively low so that some clinical trials may not have been powered to test these hypotheses [11]. Our data showing the impact of p16 using isogenic cell line models with proficient pRb is therefore relevant to address this question and encourage the scientific community to interrogate p16 using different cut-off points and along with other biomarkers of CDK4/6i resistance.

In summary, we have added the following statement, as suggested by the reviewer: “The identification of p16 overexpression as biomarker of resistance to CDK4/6i is in contrast to the results observed in PALOMA-2 and MONALEESA-2 and could be due to the specific cut-offs applied or the specimens utilized [10, 12].”

Reviewer #2 (Remarks to the Author):

In this study, the authors aim to identify biomarkers of resistance to CDK4/6 inhibition (ribociclib) in a panel of 37 patient-derived tumor models, using genetic, transcriptomic and proteomic approaches. The study claims that 1) high p16 protein levels and heterozygous RB1 loss as novel biomarkers for resistance to CDK4/6i treatment and suggest that 2) CDK4/6i plus PI3Ki may be effective in non-basal-like tumors that progress to CDK4/6i and ET, independently of the PIK3CA or RB1 status. There are several novel findings in this work including that the highest expression levels of p16 may denote CDK4/6 resistance. The authors also propose the integrated biomarker of p16 high, pRb low, cyclin D1 high, and cyclin E1 high expression which showed higher sensitivity (87%) and accuracy (85%) for the detection of ribociclib-resistant models compared to single or binary biomarkers. The combination of PI3CA and CDK4/6 inhibition is not novel and has been described and is already in clinical trials. However, this work does offer rationale for integration into the post-CDK4/6 space, elimination of endocrine backbone and de-escalation strategies which could be translationally relevant.

We thank the reviewer for highlighting the novel aspects of our study.

1) PAM50 analysis of 23 PDX models identified 3 basal-like subtypes which were all resistant to ribociclib which is expected. However, this does highlight continue translational relevance of PAM50 which is not currently used in standard practice for early or late stage breast cancer.

PAM50 identified three ER+ PDXs that were basal-like, whose CDK4/6i response clustered together with the three TNBC PDXs included in this study. As pointed by the reviewer, this data confirms the translational relevance of the PAM50 biomarker. Even though PAM50 is currently not used in standard clinical practice, various clinical trials have demonstrated its clinical utility for improved subtype classification and prediction of therapy response. To strengthen this point, we have modified the Discussion:

“Of note, PAM50-based intrinsic subtyping has become a potential indicator of benefit to CDK4/6i and our data highlight continue translational relevance of PAM50 which is not currently used in standard practice for early or late stage breast cancer [14-16].”

2) In this analysis, higher protein levels of p16 levels ($p=0.01$) were detected in resistant PDXs compared to the sensitive ones (Figure 2B). These findings were also validated in a clinical cohort of patients treated with neoadjuvant abemaciclib in the ABC-POP trial ($p=0.008$, Figure 5C). Preclinical overexpression of p16-mediated resistance to CDK4/6 inhibitors in the absence of Rb has been described previously, though this analysis is much more in depth and definitive [1]. The authors should mention that P16 protein expression has also been evaluated large scale biomarker analyses of phase III clinical trials of CDK4/6 inhibitors such as PALOMA 2. In this analysis, P16 IHC was evaluated by Ventana E6H4 anti-p16 antibody using H-score methodology of ≥ 1 as positive. A total of 568 of 666 (85%) baseline tumor tissues from the 666 patients randomized were collected. CDK6, cyclin D, cyclin E, RB, and p16 expression levels do not predict the benefit of palbociclib in combination with letrozole by IHC, FISH, or gene expression analysis [2]. This analysis reports methods of semiquantitative analysis for pRb using H-score methods though cut offs of what is considered high are not clearly presented in the manuscript. High p16 means expression score $\geq 3+$. The vast differences in cut point may partially explain why p16 in clinical trial analysis has not been linked to sensitivity or resistance to CDK4/6 inhibition and could be highlighted.

We acknowledge that p16 protein expression has been evaluated in phase III clinical trials and ask the Reviewer to read the Answer we provided to Reviewer 1, Question 5.

We also apologize for the lack of clarity regarding the cut-offs in pRb, p16 and cyclin E1/D1 employed to test their relationship with the response to ribociclib in the PDX models. In the statistical analysis section, we mentioned that: *“The optimum cut-off points established in this study were selected by the Youden index, which maximizes the sum of the sensitivity and specificity in each biomarker analyzing the ROC curve”*. We have also specified this information in the legend from Figure S2C.

3) An interesting analysis performed was assessing the value of the integrated biomarker p16 high, pRb low, cyclin D1 high, and cyclin E1 high expression. This composite marker showed higher sensitivity (87%) and accuracy (85%) for the detection of ribociclib-resistant models compared to single or binary biomarkers (Figure 2C). Validation of this integrated biomarker compared to high p16 alone in clinical samples is a logical next step.

To address this question, we performed a multivariable analysis of the effect of combining continuous p16, pRb, cyclin D1, and cyclin E1 expression values in a multivariable logistic model of KI67-response (defined as $\ln(\text{KI67}) < 1\%$ at surgery) in ER+/ HER2-negative patients treated with short-term abemaciclib in the ABC-POP trial. We have added the following statement and Figure: *“In the multivariable analyses of all patients and of patients with luminal B tumors, high p16 expression alone was significantly associated with a decreased probability of KI67-response status ($p=0.004$ and $p=0.034$ respectively) and we did not identify independent prognostic association of the other three biomarkers (Figure 5C and S4B).”* Please find the analysis for Luminal A included in the Figure 2 for Reviewers, as also requested by Reviewer 3.

In the clinical samples from patients treated with CDK4/6i in the metastatic setting, p16 high stands alone as most-significant biomarker along with pRb. Even though we included 7 additional samples that corroborate our results, the multivariable analysis was not feasible in this study due to the limited number of samples. Interestingly, the three CDK4/6i-resistant patients with high p16 levels did not exhibit low pRb. These observations have been included in the Results section.

4) 432: *“The PI3K inhibitor alpelisib sensitizes non-basal like BC PDX to ribociclib”*. This data is unfortunately not novel and has been shown before. O’Brien et al have shown that direct loss of Rb or loss of dependence on Rb signaling confers cross-resistance to inhibitors of CDK4/6, while PI3K/mTOR signaling remains activated. Treatment with alpelisib completely blocked the progression of acquired CDK4/6 inhibitor-resistant xenografts in the absence of continued CDK4/6 inhibitor treatment in models of both PIK3CA mutant and wild-type ER+/HER2– breast cancer. They also showed that triple combination therapy against PI3K:CDK4/6:ER prevented and/or delayed the onset of resistance in treatment-naïve ER+/HER2– breast cancer models [3]. This body of work does give rationale to omission of endocrine therapy in this combination and most importantly de-escalation strategies.

We thank the reviewer for highlighting the study by O’Brien et al. According to his/her comments, we have reworded the final paragraph of the Discussion as follows, to strengthen the added value of our study:

“Here, we report that the combination of ribociclib with the PI3K inhibitor alpelisib (or palbociclib with GDC-0032) has remarkable antitumor activity in non-basal like PDXs independent of PIK3CA mutation, extending the evidence from cell line-derived xenograft models [17-19]. Of note, PAM50-based intrinsic subtyping has become a potential indicator of benefit to CDK4/6i and our data highlight continue translational relevance of PAM50 which is not currently used in standard practice for early or late stage breast cancer [14, 16]. In addition, we demonstrated antitumor efficacy of the triple combination (CDK4/6i plus PI3Ki and ET) in PDXs harboring RB1 mutation, which may be an appropriate first line treatment strategy for patients harboring heterozygous RB1 loss [20]. Finally, we provide rationale for de-escalation strategies, either by omission of endocrine therapy or reduction of the PI3Ki/CDK4/6i dose.”

5) 435: Currently, this approach is further being tested in the context of tumors that progress after being treated with CDK4/6i-containing regimens (e.g. BYLieve, NCT03056755). This trial has preliminary results showing that the proportion of patients without disease progression at 6 months (CBR) was 50.4% showing clear efficacy of PIK3CA inhibitors post-CDK4/6 [4].

We thank the reviewer for highlighting that the BYLieve study has reported preliminary data. We have updated the statement as follows, that includes recent updates from SABCS 2021:

“Moreover, preliminary results from the BYLieve clinical trial has shown that the proportion of patients treated with alpelisib plus fulvestrant without disease progression at 6 months (CBR) was 50.4%, showing clear efficacy of PI3K- α inhibitors post-CDK4/6i and that this benefit is maintained

regardless of the duration of response to the previous CDK4/6i-based treatment or presence of CDK4/6i resistance genes in circulating tumor DNA (NCT03056755, [21-23]).”

6) 451: “Because of concerns regarding the safety of the combination of alpelisib plus ribociclib, we conducted de-escalation experiments showing that dose reduction of either ribociclib or alpelisib resulted in similar antitumor activity as the full dose tested in both PDX039 and PDX191 models. Dose reduction of both drugs, however, resulted in an attenuated efficacy”. Given that triplet preclinical data has been shown before and is already in clinical trials, I would consider elaborating on this data in the manuscript and highlighting in abstract and or discussion as toxicity of these combinations is a very significant concern. This would strengthen novel translational relevance.

We thank the reviewer for highlighting the novelty of our work regarding de-escalation strategies and have highlighted this in the abstract and discussion:

Abstract: “Combination of CDK4/6i ribociclib with PI3K inhibitor (PI3Ki) alpelisib showed antitumor activity in ER+ non-basal-like BC PDX, independently of PIK3CA or RB1 mutation, including in drug de-escalation experiments or omitting ET.”

Discussion: see question #4 from the same Reviewer

7) 501-507: “Although CDK4/6i are usually administrated in combination with ET, abemaciclib can be used as a single agent for the treatment of patients with ER+ /HER2-negative metastatic BC that progressed on ET and chemotherapy. Also, the data from the TREnd trial suggested that CDK4/6i combined with ET and CDK4/6i monotherapy had similar clinical benefit rates and overall response rates in post-menopausal women with advanced ET-pretreated ER+/HER2-negative BC. In this sense, our data demonstrates that targeting the CDK4/6 axis is a valid strategy when other targeted therapies against ER or HER2 have failed.” Unclear the direction of this statement. Single agent CDK4/6 inhibition has very little clinical relevance at this point. The TREND trial was quite small and the endocrine therapy backbone continued was that of which patients’ were already resistant to. The success of targeting the CDK4/6 axis after endocrine or trastuzumab resistance has been repeatedly shown in preclinical and clinical trials and not this body of work. I suspect you mean to offer rationale for PIK3CA/CDK4/6 inhibition without ET.

We apologize for the lack of clarity in this paragraph. Our intention was to dissect CDK4/6i-specific mechanisms of sensitivity and not to offer rationale for PI3Ki/CDK4/6 at this point. We have reworded this paragraph as follows:

“Although CDK4/6i are mostly administrated in combination with ET, abemaciclib has shown activity in ER+/HER2-negative metastatic BC that progressed on ET and chemotherapy [24]. Also, palbociclib has shown a signal of clinical activity in post-menopausal women with advanced ER+/HER2-negative BC having received prior ET [25]. These data with abemaciclib and palbociclib monotherapy support the examination of CDK4/6i monotherapy-specific mechanisms of sensitivity. We observed that CDKN2A/B were more frequently disrupted in ribociclib-sensitive models and the best PDX responder lacked expression of p16. Similarly, in the phase I trial of abemaciclib the best BC responder harbored a concomitant deletion of CDKN2A and CDKN2B in her tumor [26]. This observation is consistent with the increased sensitivity to palbociclib observed in 544 cancer cell lines in association with CDKN2A inactivation by DNA methylation [27] and with the limited capacity of current CDK4/6i to inhibit its target when it is bound to INK4 proteins or to p27^{Kip1} [8, 9, 28].”

Reviewer #3 (Remarks to the Author):

This manuscript employs the power of multiple PDXs to preclinically ask several questions about CDK4/6i. The authors first ask what biomarkers predict resistance to ribociclib, and identify four. RB and Cyclin E have already been reported to be correlated with CDK4/6i. This manuscript confirms these data and adds p16 and cyclin D1 expression. Of these, p16 is validated in cohort studies. The authors then move on to look at RB heterozygosity, I am unsure of what data prompted this. In a tumor with RB heterozygosity, variants with additional mutations showed altered drug sensitivity. This finding is validated in a cohort although its incidence is unclear. Finally, the authors ask if a PI3K inhibitor will add to a CDK4/6i, with or without ER inhibition, and show interesting *in vivo* data. Strengths of this paper include a panel of 21-23 PDXs, and experiments that investigate clinically important questions. For almost all of the experiments weaknesses exist as listed below. These include a very small number of resistant models, inclusion of triple-neg PDXs which were never expected to matter, inadequate functional experiments, lack of attention to ER mutations, and inadequate discussion of pharmacokinetic variables. If thoroughly fixed, the manuscript may inform future clinical trials.

We thank the reviewer for providing positive feedback.

1. Of the initial 21 PDXs, very few were sensitive to ribociclib (1 CR, 2 PR, 2 SD). This does not supply a robust “N” to distinguish molecular characteristics of responders vs nonresponders. Do the trends hold up if SD is eliminated?

We agree with the reviewer that comparing 5 responders vs. 18 non-responders provides limited statistical power. Nonetheless, the trend on p16 ($p=0.03$) holds up even if the two SD models are eliminated. Please see Figure 1D for reviewers.

2. Why were the basal PDXs included?

We included TNBC models to provide evidence that both ER+/- models with a basal-like profile by PAM50 and TNBC behaved similarly, namely were resistant to CDK4/6i.

This is important since there is clinical evidence showing that the PAM50 subtype classification is informative with regards to CDK4/6i sensitivity:

1. In the NeoPalAna, resistance to palbociclib plus anastrozole was associated with non-luminal subtypes by PAM50-based intrinsic subtype assignment [14].
2. Correlative biomarker analysis across the MONALEESA trials, showed that all subtypes except basal-like had significant PFS benefit with addition of ribociclib [15].

3. Data in Fig 2D should be confirmed with an independent cohort such as METABRIC.

Figure 2D is now Figure S2B, where we show the relationship between p16 high and pRb low, as well as pRb low and cyclin E high.

The METABRIC public dataset exclusively has mRNA data. We confirmed the relationship between pRb low and cyclin E high in METABRIC, shown as figure S2C.

4. The p16 functional data should be performed with two cell lines. Both functional investigations must include *in vitro* and *in vivo* data.

We have now included the p16 functional data for the MCF7 cell line, in addition to the T47D cell line data in Figure 4A-C and S3. As with T47D, we have observed an increase in IC50 for fulvestrant, ribociclib or the combination of drugs in MCF7 cells overexpressing p16 (Figure 4A). MCF7-p16-overexpressing cells were also enriched overtime after treatment with ribociclib *in vitro* (Figure 4C). The biochemical analysis revealed that ribociclib treatment resulted in CDK6 upregulation in T47D cells while p16-overexpression resulted in reduced CDK6 expression in MCF7 cells, so that we cannot exclude that CDK6 modulation contributes in part to the CDK4/6i

resistance phenotype observed in T47D cells *in vitro* (Figure 4B). We have amended the Results section to include this information.

In addition, we conducted *in vivo* experiments with T47D and MCF7 cells overexpressing p16. The results are shown in Figure for Reviewers 4. In the time frame that the experiments were feasible before hitting toxicity (due to estrogen pellets in NSG mice), we did not observe that p16-overexpression resulted in resistance to ribociclib. In addition, p16 levels achieved by doxycycline induction *in vivo* were 85% to 50% lower than in the *in vitro* setting, which may have limited the validation of p16 as resistant biomarker *in vivo*. Moreover, we did not observe increased CDK6 in T47D-treated cells *in vivo* which may also have contributed to the lack of CDK4/6i resistance phenotype *in vivo*. Noteworthy, we observed a general induction of CDK6 but not CDK4 upon treatment with ribociclib in our PDX panel which may represent a general adaptive response to treatment with CDK4/6i (Figure for Reviewers 4D).

We would kindly like to remind the Reviewer that in the multivariable analyses of the ABC-POP trial, high p16 expression was significantly associated with a decreased probability of Ki67-response ($p=0.004$; Figure 5C and Figure S4B), validating the observation being made in PDXs and in isogenic cell lines *in vitro*.

5. In Fig. 5C, can the data be broken out by molecular subtype?

The analysis shown in Figure 5C is now split for all tumors and for Luminal B tumors. We provide the data for Luminal A tumors as Figure for Reviewers 2A. While high p16 levels were associated with lack of response in all tumors and in Luminal B tumors, p16 did not predict treatment benefit in Luminal A tumors. This was not unexpected, since the majority of the Luminal A tumors (29 out of 33, 88%) were sensitive to CDK4/6 inhibition in terms of drop of Ki67 at day 15 of treatment.

6. The metastatic data in Fig 5D, while significant, is based on 2 resistant tumors. This major limitation should be added to Results.

We have added 7 patient samples so that we compare 6 resistant patients to 11 sensitive patients. The results for p16 are statistically significant. We have added the patient numbers in the Results section:

“Next, we tested if high p16 was also associated with lack of response to abemaciclib as single agent in the metastatic setting (n=17). Higher p16 levels ($p=0.04$) and a trend towards high cyclin E1 levels ($p=0.03$) were detected in resistant tumors (n=6) compared to the sensitive ones (n=11), whereas levels of pRb and cyclin D1 were not associated with CDK4/6i resistance as expected (Figure 5D).”

7. It is unclear how the transition from p16 to RB heterozygosity occurred. Fig 6C must list the number of specimens with RB heterozygous loss.

We apologize for the lack of clarity explaining the transition from p16 to RB heterozygosity. We have reworded the beginning of the paragraph as follows:

“We next investigated mechanisms of acquired resistance in PDXs and posited that tumors with an underlying RB1 heterozygous loss tend to acquire RB1 point mutations that result in CDK4/6i resistance. This hypothesis was based on the evidence from one patient with matched pre/post-CDK4/6i tumor sample and the relatively low frequency of acquired RB1 point mutations after treatment with palbociclib [5, 6, 20].”

Regarding the identification of specimens with RB1 heterozygous loss, please see the upper legend in Figure 6A. We have now modified the legend so that it matches the terminology of text (homo/heterozygous loss as equivalent to deep/shallow deletion).

8. The dose de-escalation study in Fig 7 should provide detail on the serum levels of the drugs, versus what is clinically achievable and safe.

Unfortunately, we did not perform serum analyses, nor collected blood samples from the dose de-escalation study so that we will answer this reviewer question based on existing data.

The highest dose in mice used in our study (75mg/kg daily, continuous) was chosen as it was described to best mimic the clinical PK in human for the 600mg/day dose [29], which is the recommended starting dose for the treatment of metastatic breast cancer (3 wks-on/1wk-off). Two dose modifications are recommended for the cases with adverse effects, namely 400 and 200 mg/day. Given that information, we conducted dose de-escalation to half the dose of ribociclib from 75 to 33 mg/kg, which would be equivalent to 300 mg/day in human. For the triplet combinations in human, few safety data is available: (i) with the pan-PI3K buparlisib and fulvestrant, the recommended phase II dose was ribociclib 400 mg/day [30]; (ii) with alpelisib plus letrozole preliminary results for the 300-500 mg/day (3 out of 4 weeks) dose have been reported [31]. In summary, the proposed dose reduction for ribociclib is within the range of doses being tested/recommended in the clinic for these combinations.

Regarding alpelisib, we proposed a dose reduction from 35 mg/kg to 15 mg/kg. According to Novartis internal reports, oral administration of alpelisib at a repeated daily dose of 25 mg/kg (plasma AUC₀₋₂₄, ss in mouse 90.65 h* μ mol/L or ~40,000 ng*h/mL) resulted in tumor regression in the Rat1-myr-p110 α tumor bearing mouse model and in MCF7 xenografts in combination with fulvestrant. At this dose, steady state exposure in mouse translates to an AUC_{0-24,ss} of ~32600 ng*h/mL in human when adjusting for the difference in protein binding. According to the dose proportionality figure shown in Figure for Reviewers 3A, this level of AUC is achieved with doses ranging 300-400mg/kg in human. Extrapolating from this information, the 15mg/kg dose used in the de-escalation study would be equivalent to 200mg/kg of alpelisib in human. Given the unexpected toxicity with buparlisib, no RP2D was established for the alpelisib combination [30] but previous safety data showed that the 200mg/kg dose is within the range of doses being tested in the clinic for these combinations [BYL (200–250 mg)][31].

We have added this statement to the Results section: “*Of note, both dose reductions were within the range of doses being tested/recommended in the clinic for these combinations [31].*”

9. The issue of ER mutation is not analyzed and is needed.

ESR1 status is provided in Figure 2A and we did not observe an association with ribociclib response, even though the representation of *ESR1*-mutant PDX was low (n=3). Nevertheless, we highlight in the text that: “*we demonstrated that ribociclib plus alpelisib (or the triple combination with fulvestrant) was effective in two PIK3CA-wt, ESR1-mut PDXs with primary resistance to ribociclib plus fulvestrant (PDX131 and PDX244LR#18R, Figure S4G and Figure 7D) or in a PDX from a patient who showed an early progression when treated with palbociclib plus letrozole (PDX450, PIK3CA and ESR1 mutant; Figure 7E).*”

We have also included this important observation in the Abstract and in the Discussion.

References

1. Serrano, M., G.J. Hannon, and D. Beach, *A new regulatory motif in cell-cycle control causing specific inhibition of cyclin D/CDK4*. *Nature*, 1993. **366**(6456): p. 704-7.
2. Shapiro, G.I., et al., *Reciprocal Rb inactivation and p16INK4 expression in primary lung cancers and cell lines*. *Cancer Res*, 1995. **55**(3): p. 505-9.
3. Dublin, E.A., et al., *Retinoblastoma and p16 proteins in mammary carcinoma: their relationship to cyclin D1 and histopathological parameters*. *Int J Cancer*, 1998. **79**(1): p. 71-5.
4. Herrera-Abreu, M.T., et al., *Early Adaptation and Acquired Resistance to CDK4/6 Inhibition in Estrogen Receptor-Positive Breast Cancer*. *Cancer Res*, 2016. **76**(8): p. 2301-13.
5. O'Leary, B., et al., *The Genetic Landscape and Clonal Evolution of Breast Cancer Resistance to Palbociclib plus Fulvestrant in the PALOMA-3 Trial*. *Cancer Discov*, 2018. **8**(11): p. 1390-1403.
6. Condorelli, R., et al., *Polyclonal RB1 mutations and acquired resistance to CDK 4/6 inhibitors in patients with metastatic breast cancer*. *Ann Oncol*, 2018. **29**(3): p. 640-645.
7. Li, Z., et al., *Loss of the FAT1 Tumor Suppressor Promotes Resistance to CDK4/6 Inhibitors via the Hippo Pathway*. *Cancer Cell*, 2018. **34**(6): p. 893-905 e8.
8. Green, J.L., et al., *Direct CDKN2 Modulation of CDK4 Alters Target Engagement of CDK4 Inhibitor Drugs*. *Mol Cancer Ther*, 2019. **18**(4): p. 771-779.
9. Li, Q., et al., *INK4 tumor suppressor proteins mediate resistance to CDK4/6 kinase inhibitors*. *Cancer Discov*, 2021.
10. DeMichele, A., et al., *CDK 4/6 inhibitor palbociclib (PD0332991) in Rb+ advanced breast cancer: phase II activity, safety, and predictive biomarker assessment*. *Clin Cancer Res*, 2015. **21**(5): p. 995-1001.
11. Finn, R., Y. Jiang, and H. Rugo, *Biomarker analyses from the phase 3 PALOMA-2 trial of palbociclib (P) with letrozole (L) compared with placebo (PLB) plus L in postmenopausal women with ER + /HER2- advanced breast cancer (ABC)*. *Annals of Oncology*, 2016. **27** (Supplement 6): p. vi552-vi587.
12. Hortobagyi, G.N., et al., *Updated results from MONALEESA-2, a phase III trial of first-line ribociclib plus letrozole versus placebo plus letrozole in hormone receptor-positive, HER2-negative advanced breast cancer*. *Ann Oncol*, 2018. **29**(7): p. 1541-1547.
13. Andre, F., et al., *Ribociclib + letrozole for first-line treatment of hormone receptor-positive (HR+), human epidermal growth factor receptor 2-negative (HER2-) advanced breast cancer (ABC): efficacy by baseline tumor markers*. *Cancer Research*, 2017. **77**(13 Supplement): p. CT045-CT045.
14. Ma, C.X., et al., *NeoPalAna: Neoadjuvant Palbociclib, a Cyclin-Dependent Kinase 4/6 Inhibitor, and Anastrozole for Clinical Stage 2 or 3 Estrogen Receptor-Positive Breast Cancer*. *Clin Cancer Res*, 2017. **23**(15): p. 4055-4065.
15. Prat, A., et al., *Correlative Biomarker Analysis of Intrinsic Subtypes and Efficacy Across the MONALEESA Phase III Studies*. *J Clin Oncol*, 2021. **39**(13): p. 1458-1467.
16. Finn, R., et al., *Comprehensive gene expression biomarker analysis of CDK 4/6 and endocrine pathways from the PALOMA-2 study*. *Cancer Research*, 2018. **78**(4 Supplement): p. P2-09-10.
17. Vora, S.R., et al., *CDK 4/6 inhibitors sensitize PIK3CA mutant breast cancer to PI3K inhibitors*. *Cancer Cell*, 2014. **26**(1): p. 136-49.
18. Jansen, V.M., et al., *Kinome-Wide RNA Interference Screen Reveals a Role for PDK1 in Acquired Resistance to CDK4/6 Inhibition in ER-Positive Breast Cancer*. *Cancer Res*, 2017. **77**(9): p. 2488-2499.
19. O'Brien, N.A., et al., *Targeting activated PI3K/mTOR signaling overcomes acquired resistance to CDK4/6-based therapies in preclinical models of hormone receptor-positive breast cancer*. *Breast Cancer Res*, 2020. **22**(1): p. 89.
20. Wander, S.A., et al., *The Genomic Landscape of Intrinsic and Acquired Resistance to Cyclin-Dependent Kinase 4/6 Inhibitors in Patients with Hormone Receptor-Positive Metastatic Breast Cancer*. *Cancer Discov*, 2020. **10**(8): p. 1174-1193.

21. Rugo, H.S., et al., *BYLieve: A phase II study of alpelisib (ALP) with fulvestrant (FUL) or letrozole (LET) for treatment of PIK3CA mutant, hormone receptor-positive (HR+), human epidermal growth factor receptor 2-negative (HER2-) advanced breast cancer (aBC) progressing on/after cyclin-dependent kinase 4/6 inhibitor (CDK4/6i) therapy.* *Journal of Clinical Oncology*, 2018. **36**(15_suppl): p. TPS1107-TPS1107.
22. Chia, S., et al., *Effect of duration of prior cyclin-dependent kinase 4/6 inhibitor (CDK4/6i) therapy (≤ 6 mo or > 6 mo) on alpelisib benefit in patients with hormone receptor-positive (HR+), human epidermal growth factor receptor 2-negative (HER2-), PIK3CA-mutated advanced breast cancer (ABC) from BYLieve.* SABCS Annual Meeting, 2021: p. Abstract P1-18-08.
23. Juric, D., et al., *Alpelisib + endocrine therapy (ET) in patients with hormone receptor-positive (HR+), human epidermal growth factor receptor 2-negative (HER2-), PIK3CA-mutated advanced breast cancer (ABC) previously treated with cyclin-dependent kinase 4/6 inhibitor (CDK4/6i): Biomarker analyses from the Phase II BYLieve study.* SABCS Annual Meeting, 2021: p. Abstract P5-13-03.
24. Dickler, M.N., et al., *MONARCH 1, A Phase II Study of Abemaciclib, a CDK4 and CDK6 Inhibitor, as a Single Agent, in Patients with Refractory HR(+)/HER2(-) Metastatic Breast Cancer.* *Clin Cancer Res*, 2017. **23**(17): p. 5218-5224.
25. Malorni, L., et al., *Palbociclib as single agent or in combination with the endocrine therapy received before disease progression for estrogen receptor-positive, HER2-negative metastatic breast cancer: TREnd trial.* *Ann Oncol*, 2018. **29**(8): p. 1748-1754.
26. Patnaik, A., et al., *Efficacy and Safety of Abemaciclib, an Inhibitor of CDK4 and CDK6, for Patients with Breast Cancer, Non-Small Cell Lung Cancer, and Other Solid Tumors.* *Cancer Discov*, 2016. **6**(7): p. 740-53.
27. Li, P., et al., *P16 methylation increases the sensitivity of cancer cells to the CDK4/6 inhibitor palbociclib.* *PLoS One*, 2019. **14**(10): p. e0223084.
28. Guiley, K.Z., et al., *p27 allosterically activates cyclin-dependent kinase 4 and antagonizes palbociclib inhibition.* *Science*, 2019. **366**(6471).
29. Kim, S., et al., *The potent and selective cyclin-dependent kinases 4 and 6 inhibitor ribociclib (LEE011) is a versatile combination partner in preclinical cancer models.* *Oncotarget*, 2018. **9**(81): p. 35226-35240.
30. Tolaney, S.M., et al., *Phase Ib Study of Ribociclib plus Fulvestrant and Ribociclib plus Fulvestrant plus PI3K Inhibitor (Alpelisib or Buparlisib) for HR(+) Advanced Breast Cancer.* *Clin Cancer Res*, 2021. **27**(2): p. 418-428.
31. Juric, D., *Phase Ib/II study of ribociclib and alpelisib and letrozole in ER+, HER2- breast cancer: Safety, preliminary efficacy and molecular analysis.* *Cancer Research*, 2016(76(4 Supplement):P3-14-01).

FIGURES FOR REVIEWERS

Figure for reviewers 1

Figure for reviewers 1. A) Relative tumor volume in 23-untreated PDX in relationship with ribociclib-response for the indicated period of time. Mean and \pm SEM are indicated. p -value, unpaired parametric t -test. **B)** Co-occurrence of altered pRb and cyclin D1 expression levels in two cohorts of ER⁺ breast carcinomas, one from the TCGA database (n=814) and other from METABRIC database (n=1817), using cBioportal (www.cBioportal.org). The cut-off for high versus

low protein/mRNA levels is indicated. OR: odd's ratio; prot: protein; SD: standard deviation. Quantification of IHC staining for p16, cyclin E1 and cyclin D1 in 23-untreated PDX in relationship with pRb expression also quantified by IHC **(C)** or IHC staining for p16, pRb, cyclin E1 and cyclin D1 in 23-untreated PDX in relationship with ribociclib-response **(D)**. Semiquantitative analysis was performed for pRb and p16, or the Allred scoring method for cyclin E1 and cyclin D1 in relationship with ribociclib-response. Different colors indicate the PDX intrinsic subtype, \$ indicates the models harboring gene amplification and different symbols indicate the response to ribociclib. Mean and \pm SEM are indicated. *p*-value, unpaired parametric *t*-test. R: resistant; S: sensitive.

Figure for reviewers 2

A

Figure for reviewers 2. A) Forest plots displaying the Odd's ratios and 95% confidence intervals (CI) for the KI67 response of Luminal A tumors to abemaciclib of the indicated biomarkers. *p*-value are also indicated.

Figure for reviewers 3

Figure for reviewers 3. A) Dose proportionality for alpelisib AUC0-24,ss of single agent in cancer subjects (PH1PAS-mono). Solid line is $AUC_{0-24} = \exp(\alpha) \cdot \text{dose}^{\beta}$ where α is the intercept (on the log scale) for non-Japanese females. Dotted reference line is $\text{aucinf} = \exp(\alpha) \cdot \text{dose}$.

Figure for reviewers 4

Figure for reviewers 4. A) Relative tumor growth of T47D-p16 and MCF7-p16 xenograft tumors after treatment with the indicated drug(s) for the indicated period of time. Dashed lines indicate the range of PD (>1.2), SD (1.2 to -0.7) and PR/CR (<-0.7). **B)** Representative pictures showing IHC staining of p16 in MOCK and p16 overexpressing T47D and MCF7 cells in vivo vs in vitro. Yellow squares mark off areas showed with bigger magnification. **C)** Immunoblot of the indicated proteins in MOCK and p16 overexpressing T47D and MCF7 cells in vivo vs in vitro. **D)** Immunoblot quantification of CDK4 and CDK6 in vehicle and ribociclib-treated PDXs. For illustration purposes,

the mean value of each PDX was plotted; however, for the statistical analysis all technical replicates were used. p -values are based on Mann-Whitney U test.

REVIEWERS' COMMENTS

Reviewer #2 (Remarks to the Author):

To the authors, it was a pleasure to read this note worthy work of which I think could have novel translational potential. There are several novel findings in this work including that the highest expression levels of p16 may denote CDK4/6 resistance. The authors also propose the integrated biomarker of p16 high, pRb low, cyclin D1 high, and cyclin E1 high expression which showed higher sensitivity (87%) and accuracy (85%) for the detection of ribociclib-resistant models compared to single or binary biomarkers. This finding could lead to validation in clinical samples. The combination of PIK3CA and CDK4/6 inhibition is not novel and has been described and is already in clinical trials. However, this work does offer rationale for integration into the post-CDK4/6 space, elimination of endocrine backbone and de-escalation strategies which could be translationally relevant. The methodology is thorough and sound. I require no further revisions.

Reviewer #3 (Remarks to the Author):

The authors attempted to explain or retest many variables that were questioned in the first submission of this paper. The preclinical data have been improved by the addition of cell lines, in vivo data and better characterization.

In looking at the likely impact of the manuscript, the p16 hypothesis presented herein is already questioned by data in two large trials. Other papers have reported portions of the preclinical data. A high profile paper on this topic would have to definitively settle the issue, which this manuscript cannot based on the size alone of the clinical component. At best, the paper will prompt investigators to include p16 in future analyses.